# Triple Andreev dot chains in semiconductor nanowires

Hao Wu,[1] Po Zhang,[1] John P. T. Stenger,[1] Zhaoen Su,[1] Jun Chen,[2] Ghada
Badawy,[3] Sasa Gazibegovic,[3] Erik P. A. M. Bakkers,[3] and Sergey M. Frolov[1, *]

[1]*Department of Physics and Astronomy, University of Pittsburgh, Pittsburgh, PA, 15260, USA*
[2]*Department of Electrical and Computer Engineering,*
*University of Pittsburgh, Pittsburgh, PA, 15260, USA*
[3]*Department of Applied Physics, Eindhoven University of Technology, 5600 MB Eindhoven, The Netherlands*

Kitaev chain is a theoretical model of a one-dimensional topological superconductor with Majorana zero modes at the two ends of the chain. With the goal of emulating this model, we build a chain of three quantum dots in a semiconductor nanowire. We observe Andreev bound states in each of the three dots and study their magnetic field and gate voltage dependence. Theory indicates that triple dot states acquire Majorana polarization when Andreev states in all three dots reach zero energy in a narrow range of magnetic field. In our device Andreev states in one of the dots reach zero energy at a lower field than in other two, placing the Majorana regime out of reach. Devices with greater uniformity or with independent control over superconductor-semiconductor coupling should realize the Kitaev chain with high yield. Due to its overall tunability and design flexibility the quantum dot system remains promising for quantum simulation of interesting models and in particular for modular topological quantum devices.

## INTRODUCTION

Quantum dots are a versatile platform for quantum computation. Quantum dot-based spin qubits show increasing gate fidelities [1]. Superconducting dots are used to manipulate the spins of Andreev states [2]. Quantum dots in a chain or array can be used to simulate Hamiltonians of complex many-body states [3, 4]. Kitaev chain is a model that describes a one-dimensional topological superconductor with p-wave pairing [5]. This pairing can be effectively engineered, for instance, by combining one-dimensional semiconductor nanowires that have strong spin-orbit interaction, with proximity-induced s-wave superconductivity, and Zeeman spin splitting [6, 7]. This is the continuous Majorana wire model that motivates the study of Majorana zero modes in hybrid superconductor/semiconductor heterostructures.

Zero-bias conductance peak, a signature of Majorana zero modes, has been observed in tunneling experiments on hybrid superconductor/semiconductor nanowire devices [8–10]. Zero-bias peaks are necessary but not sufficient to establish Majorana modes. Andreev bound states in quantum dots also produce zero-bias peaks in the same devices [11–13]. Andreev bound states appear generically in superconductor/semiconductor nanowire systems due to disorder or device geometry [14–17]. Conventional wisdom has been that these trivial Andreev bound states are detrimental to Majorana experiments.

In this work, rather than trying to eliminate Andreev states, we take advantage of them in our Majorana search. Theory suggests a robust topological superconducting phase in chains of Andreev quantum dots [18, 19]. A one-dimensional system is assembled out of multiple dots coupled to superconductors, each with an Andreev bound state. Because sites are independently tunable, it helps to suppress and overcome the effect of disorder.

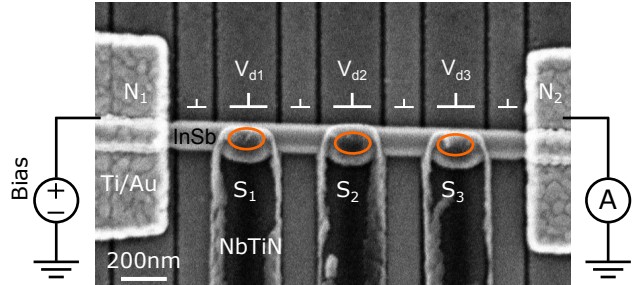

FIG. 1. Scanning electron micrograph (SEM) of the triple-dot device. Orange circles indicate positions of quantum dots within the InSb nanowire, underneath three superconducting leads ($S_1$, $S_2$, $S_3$, NbTiN), electrostatic gates that separate and tune dots are marked with inverted T's. The circuit shown is the primary measurement configuration where non-superconducting leads $N_1$ and $N_2$ are the source and drain.

While we are motivated by early theoretical proposals, we do not aim to replicate them and instead develop a theory tailored to our devices (see supplementary information for discussion).

We build a device of three quantum dots in a chain in an InSb nanowire. At zero magnetic field, Andreev bound states exist in all three dots. Andreev bound states from each dot are gate-dependent. No other unwanted or uncontrollable dots are found in the main wire segment. We observe dot-dependent zero-bias conductance peaks in magneto-transport spectroscopy. We interpret zero-bias peaks as Andreev states along the triple-dot chain crossing zero energy at finite magnetic fields. We build a tight-binding model of a quantum dot chain containing three sites. We simulate the energy spectrum and transport of the triple-dot chain. For such a short chain, within a narrow parameter window, the probability distribution of Majorana wavefunctions indicates a

partial separation of two Majorana zero modes localized at two end sites (see supplementary materials for simulation results).

In our device, transport is dominated by one of the quantum dots in the chain, $D_3$, which has a zero-bias crossing at lower magnetic fields than the other two dots. Thus the parameters lie outside of the simulated non-trivial topological phase. The device geometry is suitable for studying the correlation of two end states with non-local measurements. Even though the desired regime is not accessible, this device with new geometry has overcome many nanofabrication challenges and is one step further towards an unambiguous identification of Majorana zero modes.

## DEVICE DESCRIPTION

Fig. 1 shows the scanning electron micrograph (SEM) of the triple-dot device. An indium antimonide (InSb) semiconductor nanowire is contacted by three separate superconductors ($S_1$, $S_2$, $S_3$) to create three Andreev quantum dots ($D_1$, $D_2$, $D_3$) in a chain. The hybrid nanowire device has two non-superconductor leads ($N_1$, $N_2$) at wire ends for probing the chain. Three quantum dots are separated along the nanowire by tunnel barriers controlled by narrower unlabeled electrostatic gates located in between contacts. Quantum dots are tuned by wide gates (d1, d2, d3). Magnetic fields parallel to the nanowire axis can be applied. Fig. 1 shows the measurement configuration used for most data acquired: we apply a voltage bias through $N_1$, and measure current and differential conductance from $N_2$ while floating $S_1$, $S_2$, and $S_3$. In this configuration current flows through all three Andreev dots, thus resonances and their positions are influenced by quantum states in the entire chain. The measurements can be configured differently utilizing other terminals if a specific section of the device is of particular interest (see Fig. 5 below).

## RESULTS

In our five-terminal, seven-gate device, we test the routine of setting up quantum dots within the chain. We understand where the dots are located, what their individual states are, and how to control them and their couplings. Fig. 2 shows representative differential conductance spectra of the dot chain while tuning each dot over several occupation values. Current passes through all three dots between $N_1$ and $N_2$. Conductance resonances are pinned to finite bias and exhibit varying degrees of wiggling but never cross zero bias at zero magnetic field (Fig. 2(a)-(c)).

In a single quantum dot, these resonances are transitions from ground to excited Andreev states: the dot

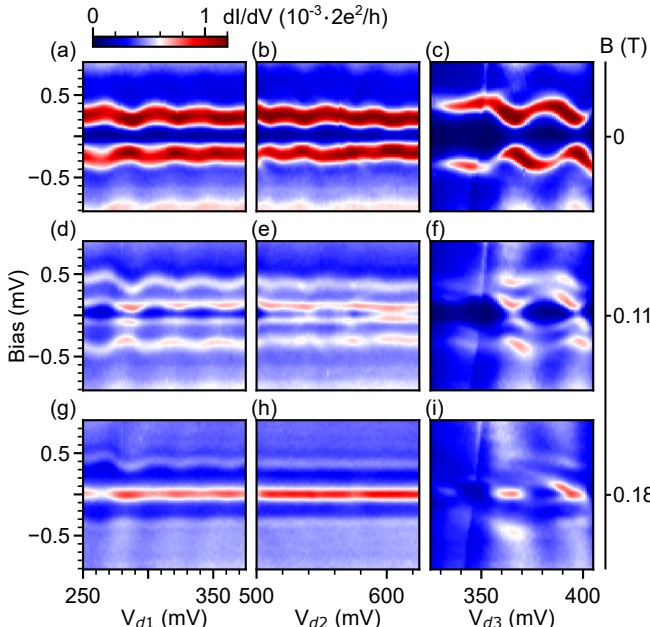

FIG. 2. Bias spectroscopy of Andreev bound states in each dot at different magnetic fields. (a)-(c): B=0, (d)-(f): B=0.11T, (g)-(i): B=0.18T. Gate settings in (a,d,g): $V_{d2}$=576.5mV, $V_{d3}$=395mV. (b,e,h): $V_{d1}$=322.5mV, $V_{d3}$=395mV. (c,f,i): $V_{d1}$=322.5mV, $V_{d2}$=576.5mV. Measured from $N_1$ to $N_2$.

is in the ground state at the start of the charge transfer cycle and the additional electron passing through the dot enters the excited state [14, 20]. Because the dots are directly underneath the leads, they are expected to be strongly coupled to superconductors, with the ground state always a singlet while the excited state a doublet, at zero magnetic field. In a triple-dot chain where current passes through all three dots, these resonances are a convolution of ground-excited state transitions in all dots. Thus we cannot unambiguously extract energy scales in each dot from these data. However, we can evaluate the effect of each dot's gate on the resonances. In Figs. 2(a,b), Andreev bound states appear more flat. We interpret this as $D_1$ and $D_2$ being stronger coupled to leads $S_1$ and $S_2$. In Fig. 2(c), Andreev states appear wavier, suggesting a weaker coupling of $D_3$ to $S_3$.

We note that the differential conductance is relatively low, less than a percent of a quantum of conductance. This is because current has to pass through four tunneling barriers. For measurements in all figures, we set the barrier between $S_3$ and $N_2$ to be the least transmitting. This is a compromise tuning strategy: raising more barriers leads to sharper resonances, but the signal decreases further.

We observe doubling the number of resonances when a parallel magnetic field is applied in Fig. 2(d)-(f), at B=0.11T. This is because the excited Andreev states (doublets) Zeeman split, while the ground state remains

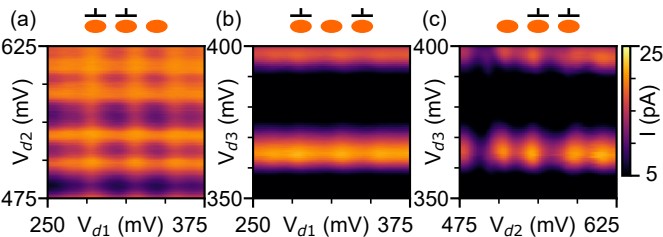

FIG. 3. Stability diagrams between (a) $D_1$ and $D_2$, $V_{d3}$=366.25mV. (b) $D_1$ and $D_3$, $V_{d2}$=537.5mV. (c) $D_2$ and $D_3$, $V_{d1}$=315mV. Measured from $N_1$ to $N_2$. Bias voltage V=0.2mV. B=0.

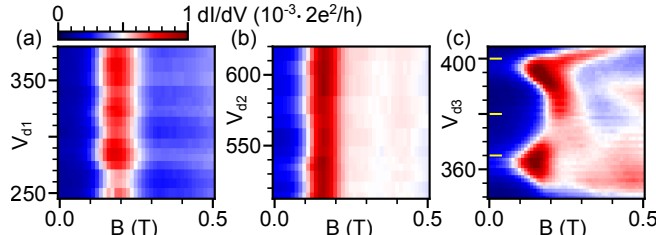

FIG. 4. Zero-bias tunneling conductance as a function of magnetic fields in (a) $D_1$, $V_{d2}$=576.5mV, $V_{d3}$=392.5mV. (b) $D_2$, $V_{d1}$=322.5mV, $V_{d3}$=360mV. (c) $D_3$, $V_{d1}$=185mV, $V_{d2}$=472.5mV. Measured from $N_1$ to $N_2$. Magnetic field evolution of the triple-dot Andreev bound states at different $V_{d3}$ values indicated by yellow short lines in (c) are in supplementary materials.

a singlet and does not split with applied magnetic field. One branch of resonances moves to lower voltage bias. The resonances reach zero voltage bias for the first time at B=0.18T (Fig. 2(g)-(i)).

Softened induced gap allows us to observe the resonance in $D_3$ at zero voltage bias even when states in $D_1$ and $D_2$ are not near zero energy. For quantum dot chain with hard induced gap, zero-bias peak only occurs when states of all dots cross zero voltage bias simultaneously at a fixed magnetic field. When sweeping the gates of dots $D_1$ and $D_2$, we observe zero-bias conductance peaks across the entire gate voltage range of these figures (Figs. 2(g,h)). In dot $D_3$, we observe zero-bias peaks at three narrow intervals of gate voltage $V_{d3}$. If we set $V_{d3}$=365mV, we have zero-bias peaks on both nanowire ends.

This is not the situation we expect when delocalized Majorana zero modes are present. We can tell this from the fact that these zero-bias peaks are sensitive to $V_{d3}$ and not to the gates ($V_{d1}$, $V_{d2}$) controlling the other dots ($D_1$, $D_2$). In the Kitaev chain regime, we expect zero-bias peaks to depend on tuning of all dots. In Figs. 2(g,h), $V_{d3}$ is set to 395mV, where one of the zero energy crossings occurs. The 'long' zero-bias peaks in Figs. 2(g,h) indicate dot gate d1 and d2 are not coupled to the state that crosses zero voltage bias at B=0.18T, which is localized at $D_3$. The states localized at $D_1$ and $D_2$ likely cross zero voltage bias at higher magnetic fields since those Andreev states are coupled stronger to leads $S_1$ and $S_2$.

To characterize the chain of dots we provide current maps in which pairs of dot gates are tuned, with three possible combinations ($D_1$-$D_2$, $D_1$-$D_3$, $D_2$-$D_3$), in Fig. 3. These data are at zero magnetic field and at a fixed bias voltage (V=0.2mV). Note that these plots show current rather than conductance, signaled by using a different colormap than in other figures. High current here is where Andreev resonances dip to lower bias, because as Fig. 2 shows, resonances never cross zero bias at zero field. Lines of high current regions form rectangular grid patterns which indicate that capacitive couplings between the dots are weak. This is because the dot gates are screened by leads $S_1$, $S_2$, $S_3$. We only observe cur-

rent maxima related to the three dots and no extra states within the central nanowire segment. Resonances related to $D_3$ dominate over resonances from $D_1$ and $D_2$, further confirming that $D_3$ is the bottleneck of transport in this device.

Fig. 4 shows extracted zero-bias tunneling conductance as a function of dot gate voltages and parallel magnetic fields. Figs. 4(a)-(c) are effectively phase diagrams of zero-bias peaks for all three dots: red regions correspond to zero-bias peaks and blue regions correspond to no peak at zero bias. In Figs. 4(a,b), the zero-bias peaks are mostly gate independent, which is in agreement with Figs. 2(g,h). Fig. 4(c) shows the onset point of the first zero energy crossing in magnetic field is strongly dependent on $V_{d3}$. The first zero energy crossing (peak center) can take place in a range of magnetic fields between 0.16T and 0.35T.

Fig. 5 shows magnetic field evolution of the triple-dot Andreev bound states to higher magnetic fields (up to 1T). Between B=0 and 0.2T the splitting of Andreev resonances and zero energy crossing are clear and high contrast. However, signals are less prominent in magnetic fields above $\approx 0.2$T. This is due to soft superconducting gap of NbTiN in finite magnetic fields. Though it is possible to observe sharp resonances to higher fields in NbTiN devices [8, 10, 21], here we have an additional constraint that our NbTiN electrodes are relatively narrow ($\approx 200$ nm) and thin ($\approx 60$ nm) in order to reduce stress that may break nanowires for devices with multiple superconducting electrodes. This can explain weakened proximity-induced superconductivity.

At fields above 0.2T, there are several additional resonances at low voltage bias. Spectrum and transport simulations predict up to 3 zero energy crossings in magnetic fields depending on parameters of the triple-dot chain (see supplementary materials). It is plausible that resonances at higher fields originate from $D_1$ and $D_2$. However, reduced contrast prevents their unambiguous identification.

Fig. 5(b) illustrates how we can use the five-contact geometry of this device to gain additional information on state localization along the chain. We apply voltage bias through $N_2$ and measure current and differential conductance from $S_3$ while floating $N_1$, $S_1$, and $S_2$. The overall differential conductances are comparable in Figs. 5(a,b), confirming that transport features observed in this paper are dominated by $D_3$. We are measuring resonances with sharp features of $D_3$ on a faint background of $D_1$ and $D_2$. Some of the resonances beyond 0.2T do not show up in panel (b). This may be a confirmation that they are due to $D_1$ and $D_2$, which are not in the path of current in panel (b). However, it may also be due to slightly more broadened features in panel (b) reducing the sensitivity to those resonances.

We also show in supplementary materials that the tunnel barrier located at the left end near $N_1$ is significantly more open compared to the right tunnel barrier at $N_2$. The inter-dot barriers are also low. Thus whenever $N_2$ is excluded from the measurement configuration the resonances broaden considerably.

## CONCLUSIONS, LIMITATIONS AND OUTLOOK

Two conclusions can be drawn from our study. First, our measurement technique can help identify where the states that generate zero-bias peaks are localized along the chain. Therefore, application of this technique in future devices will lead to a successful identification of the Kitaev chain regime.

Second, the devices we made here - despite large degree of control over Andreev states in multiple dots - are still off from the Kitaev chain regime. Superconducting gap surviving to higher magnetic fields and further optimized coupling between the nanowire and the superconductor are desired. The limit to which we could push these measurements in the direction of the Kitaev chain mirrors efforts to realize Majorana modes in continuous 'bulk' nanowires, and faces similar challenges.

More nuanced discussion of limiting aspects follows. Tunnel barrier gates have steep gate traces making it harder to tune the chain in a balanced way, as current through the chain is sensitive to all barrier gates, and the same for inter-dot couplings. Leaving some barriers open limits the ability to use all five contacts of the device as probes, since some probes do not have high barriers between them. In future devices reducing the width of barrier gates can help by reducing gate lever arms, and also increasing the maximum possible inter-dot coupling.

Another limitation of our design is that the coupling between dot and superconductor is fixed. Due to weaker coupling between $D_3$ and $S_3$, Andreev states in $D_3$ behave differently in magnetic fields. An important regime to study in the future is where zero energy crossings of all three dots happen closely in magnetic fields. This

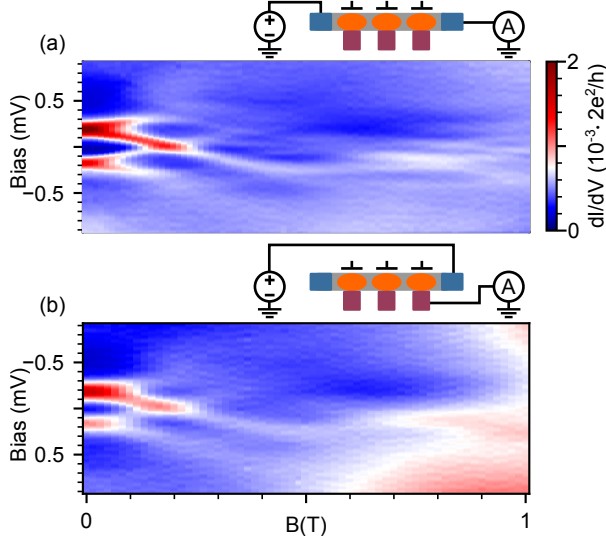

FIG. 5. Magnetic field evolution of the triple-dot Andreev bound states with different measurement configurations shown in circuit diagrams above panels, (a) $N_1$-$N_2$, (b) $N_2$-$S_3$. $V_{d1}$=315mV, $V_{d2}$=537.5mV, $V_{d3}$=396mV. Because the source and drain are reversed in (b) compared to (a), we have flipped voltage bias axis in (b).

requires either identical dots or larger tunability of dot-superconductor coupling. Intuitively, we want maximal coupling of quantum dots to superconductors in order to have the largest induced gap such that parameter window for hosting topological phase is maximized. But in practice, it may be beneficial to have a fixed or a tunable tunnel barrier between the semiconductor and the superconductor. In supplementary materials, we show another device design with triangular NbTiN electrodes where dots are defined to the side of contacts, as a way to control dot-superconductor coupling.

Collectively, these limitations make clear that within this design and fabrication method realizing chains longer than 3 dots is increasingly challenging. Well-separated Majorana zero modes are not expected for a chain of three dots outside of fine-tuned regimes, i.e. when Andreev states from all three dots reach zero bias at the same magnetic field (see supplementary materials for simulation results).

On the other hand, realizing Majorana zero modes with a few quantum dots may be less challenging than a simple chain model suggests. The total length of our triple-dot device is more than $1\mu$m, which would be sufficient to separate Majorana modes in a continuous nanowire with a 'bulk' topological phase [11, 22, 23]. The length of each dot is $\approx 200$ nm, comparable to spin-orbit-interaction length of InSb [24, 25]. Each dot can be treated as one short wire section, in which Andreev bound states can be partially separated into Majorana

modes [26, 27]. Several models already considered multiple segments of topological superconductor that couple through tunnel gates [28–31], which hybridizes inner Majorana modes on each dot leaving the outer ones unpaired, similar to the original Kitaev proposal [5].

Advances in materials synthesis and improvements in processing and nanofabrication may help resolve some issues discussed above. In-situ techniques for superconductor deposition can achieve uniform contacts and cleaner superconductor-semiconductor interfaces. Selective-area grown (SAG) nanowires with carefully designed shadow wall structures present a promising platform for scaling up to longer chains [32]. Superconducting materials with large and hard gap can be used, such as Sn [33] or Pb [34].

## FURTHER READING

Background on Majorana zero modes in hybrid superconductor-semiconductor nanowires can be found in [35–38]. Experiments on multi-terminal nanowire devices can be found in [32, 39–41]. Proposals of implementing the Kitaev chain with quantum dots and superconductors can be found in [18, 19, 42]. Studies of Andreev bound states in hybrid superconductor-semiconductor nanowire devices are discussed in in [13, 14, 20, 43]. Prospects of implementing Majorana-based quantum computation are assessed in [30, 44–48].

## EXPERIMENTAL METHODS

Nanowire growth: InSb nanowires are grown by metalorganic vapor phase epitaxy (MOVPE). A typical nanowire is 3-5 $\mu$m in length with a diameter of 120-150 nm.

Nanowire deposition: InSb nanowires are transferred onto undoped Si substrates with pre-patterned gate electrodes using a micromanipulator under an optical microscope. The gates are 1.5/6 nm Ti/PdAu covered by 10 nm of HfO$_x$ as the dielectric layer.

Contact deposition: contact patterns are written by standard electron-beam lithography. Sulfur passivation and Ar sputter cleaning are performed to remove the native oxidation layer on nanowires before sputter deposition 5/60 nm of NbTi/NbTiN at an angle of 45 degree with respect to the substrate. Sulfur passivation is performed to remove the oxidation layer before e-beam evaporation of 10/145 nm of Ti/Au.

Measurement: measurements are performed in a dilution refrigerator at a base temperature of 40 mK. Multiple stages of filtering are used to enhance the signal-to-noise ratio. All voltage bias data are two-terminal measurements. Series resistance was taken into account in calculating conductance in all figures.

## END STATEMENTS

*Data availability* Curated library of data extending beyond what is presented in the paper, as well as simulation and data processing code are available on Zenodo. [49].

*Author contributions* G.B., S.G. and E.B. provided InSb nanowires. H.W., P.Z., Z.S. and J.C. fabricated the devices. H.W., P.Z. and Z.S. performed the measurements. J.S., H.W., and P.Z. carried out simulations. H.W., P.Z., Z.S., J.S., and S.F. analyzed the data. H.W. and S.F. wrote the manuscript with contributions from all of the authors.

*Acknowledgements* The authors thank D. Pekker for fruitful discussions and M. Hocevar for technical assistance.

*Funding* S.F. is supported by NSF PIRE-1743717, NSF DMR-1906325, ONR and ARO. E.B. is supported by the European Research Council (ERC HELENA 617256), the Dutch Organization for Scientific Research (NWO), the Foundation for Fundamental Research on Matter (FOM), and Microsoft Corporation Station-Q.

———————

* frolovsm@pitt.edu

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

# BACKGROUND

## Context: Quantum Computing

Quantum dot systems are a versatile platform for quantum information processing. In quantum dot-based spin qubits, initialization, manipulation, logic gate control and readout of spin states and qubit entanglement have been demonstrated [50–54]. With high measurement and control fidelity, quantum dot-based qubit becomes a great candidate for scaling up quantum computation above the threshold of fault-tolerance [55, 56].

Quantum dots in a chain or array can be used to simulate 1D or 2D Hamiltonians of complex many-body states. It is an alternative approach to tackle hard quantum problems with strong interactions that cannot be numerically calculated with classical computers. Electrons confined in artificial atom lattice sites (quantum dots) are governed by the same physics as those in crystalline lattices. By designing a quantum dot array with desired charging energy and gate-tunable inter-dot tunnel coupling, it can mimic the quantum behaviors of electrons in materials with exotic electronic and magnetic properties and provide insights into the underlying physics. Several models have been simulated by semiconductor quantum dots including strongly correlated Fermi-Hubbard model [3] and Nagaoka ferromagnetism [4].

Issues of decoherence and scalability obstruct the realization of practical quantum computation. In order to accomplish error-correction, many physical qubits are needed to create one logical qubit [57, 58]. Topological quantum computing is an approach to circumvent the harsh requirements for logical qubits and scale up to realize fault-tolerant quantum computation [44, 59, 60]. Topological quantum computations are carried out by performing braiding operations on non-Abelian anyons, which are stable and resilient to decoherence. Although its non-Abelian exchange statistics have not been confirmed in experiments yet, Majorana zero modes in superconductor/semiconductor nanowire hybrid systems have become one of the most promising candidates for topological qubits in the past decade [8, 9, 47].

## Previous work on Majorana zero modes in superconductor/semiconductor nanowires

Kitaev chain describes a one-dimensional topological superconductor with p-wave pairing [5]. P-wave pairing, which is not readily available, can be effectively engineered, for instance, by combining one-dimensional semiconductor nanowires that have strong spin-orbit interaction, with proximity-induced s-wave superconductivity, and Zeeman effect [6, 7]. This is the continuous Majorana wire model that motivates the study of Majorana zero modes in hybrid superconductor/semiconductor heterostructures.

Zero-bias conductance peak, a signature of Majorana zero modes, has been observed in tunneling experiments of hybrid superconductor/semiconductor nanowire devices [8–10]. Zero-bias peaks are necessary but not sufficient to conclude Majorana.

Recent progress on Majorana zero modes in nanowires is mostly in material development and understanding of other origins of the zero-bias peaks. A lot of techniques have been developed to improve the quality of hybrid devices, such as the epitaxial growth of superconductors [61, 62], in-plane selective area nanowire growth [63, 64], and shadow lithography [32, 65, 66]. Andreev bound states of quantum dots can mimic signatures of Majorana zero modes including quantized zero-bias peak [11–13]. With current device quality, Andreev bound states exist commonly in hybrid superconductor/semiconductor nanowires due to disorder or device geometry [15–17]. They have also been studied intentionally in semiconductor quantum dots coupled to superconductors [14, 20, 43, 67–69]. All reported Majorana signatures can be explained by trivial Andreev bound states. Future experiments will aim to distinguish the two phenomena in a clear way through consistent demonstration of multiple Majorana signatures within the same nanowire.

## MODEL DESCRIPTION

The system is described by a spinful Hamiltonian with nearest dot coupling $t_i$, dot potentials $\mu_i$, an external magnetic field $B$, spin-orbit coupling $\alpha_i$, induced superconductivity $\Delta_i$, and electron interactions $U_i$.

$$
\begin{aligned}
H = & \sum_{i,\sigma} \mu_i d_{i,\sigma}^\dagger d_{i,\sigma} + \sum_{i,\sigma} t_i \left( d_{i+1,\sigma}^\dagger d_{i,\sigma} + h.c \right) \\
& + B \sum_i \left( d_{i,\uparrow}^\dagger d_{i,\downarrow} + h.c. \right) \\
& + \sum_i \alpha_i \left( d_{i,\uparrow}^\dagger d_{i+1,\downarrow} - d_{i,\downarrow}^\dagger d_{i+1,\uparrow} + h.c. \right) \\
& + \sum_i \Delta_i (d_{i,\downarrow}^\dagger d_{i\uparrow}^\dagger + h.c.) + \sum_i U_i d_{i,\uparrow}^\dagger d_{i,\uparrow} d_{i,\downarrow}^\dagger d_{i,\downarrow}
\end{aligned}
\tag{S1}
$$

where $i \in \{1,2,3\}$ runs over the dots, $\sigma \in \{\uparrow, \downarrow\}$ runs over the spin and $d_{i,\sigma}^\dagger$ ($d_{i,\sigma}$) creates (destroys) an electron on dot $i$ with spin $\sigma$. Let $E_n$ and $|n\rangle$ denote the eigenvalues and eigenvectors of the Hamiltonian. Majorana operators can be defined by

$$
\begin{aligned}
d_{i,\sigma}^\dagger &= \frac{1}{2} \left( \gamma_{x,i,\sigma} - i\gamma_{y,i,\sigma} \right) \\
d_{i,\sigma} &= \frac{1}{2} \left( \gamma_{x,i,\sigma} + i\gamma_{y,i,\sigma} \right)
\end{aligned}
\tag{S2}
$$

To calculate the current we allow electrons to tunnel through the ends of the devices (i.e. to and from the leftmost and rightmost dots). We assume that the electron distribution in either lead is in equilibrium described by the Fermi-distribution $f(E)$ with a potential bias of $eV$ on the left and $-eV$ on the right. We take the transition rates to be

$$
\begin{aligned}
\Gamma_{gain,nm}^{L,R} &= f(E_n - E_m \mp eV) \sum_\sigma t_{L,R}^2 |\langle n|d_{1,\sigma}^\dagger|m\rangle|^2 \\
\Gamma_{loss,nm}^{L,R} &= (1 - f(E_n - E_m \mp eV)) \sum_\sigma t_{L,R}^2 |\langle n|d_{1,\sigma}|m\rangle|^2
\end{aligned}
\tag{S3}
$$

where $t_L$ is the coupling to the left lead, $t_R$ is the coupling to the right lead, and $\Gamma_{gain,nm}^{L,R}$ is the rate at which an electron comes into the system from the right lead $R$ or left lead $L$ and excites the system from the state $|m\rangle$ to the state $|n\rangle$, while $\Gamma_{loss,nm}^{L,R}$ is the rate at which an electron leaves the system going into the right $R$ or left $L$ lead causing decaying the system from $|m\rangle$ to $|n\rangle$. Notice that we have made no assumption about which state ($|m\rangle$ or $|n\rangle$) has more electrons. The total transition rate from $|m\rangle$ to $|n\rangle$ is then given by

$$
\Gamma_{nm} = \Gamma_{gain,nm}^L + \Gamma_{loss,nm}^L + \Gamma_{gain,nm}^R + \Gamma_{loss,nm}^R
\tag{S4}
$$

We assume the system is in a steady state so that the probabilities in each state is unchanging in time

$$
0 = \frac{dP_n}{dt} = \sum_m M_{nm} P_m
\tag{S5}
$$

where

$$
M_{nm} = \Gamma_{nm} - \sum_l \delta_{nm} \Gamma_{ln}
\tag{S6}
$$

Solving the set of linear equations in Eq. S5 and using the fact that $\sum_n P_n = 1$ allows us to solve for the probabilities. From here, we can calculate the current

$$
\begin{aligned}
I = & \\
& \sum_n P_n \sum_m \left( \Gamma_{gain,mn}^L + \Gamma_{loss,mn}^R - \Gamma_{loss,mn}^L - \Gamma_{gain,mn}^R \right)
\end{aligned}
\tag{S7}
$$

**SIMULATIONS RESULTS**

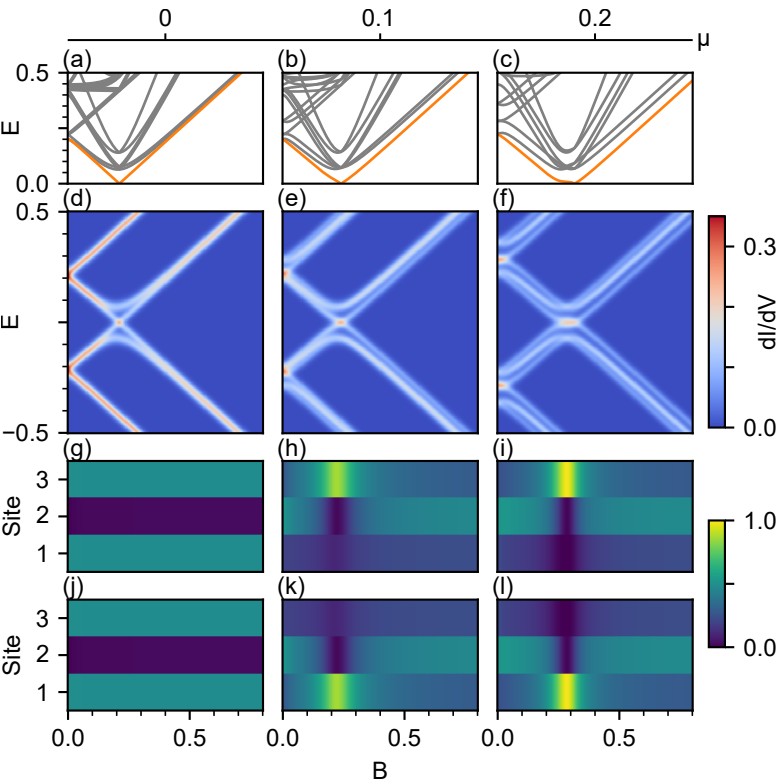

FIG. S1. Simulation results for a chain of 3 identical dots at different dot potentials, $\mu = 0, 0.1, 0.2$. (a-c) Many-body spectra versus the magnetic field. The orange line is the first excited state. (d-f) Differential conductance in magnetic field. (g-l) Evaluation of Majorana components (probability amplitudes) in different sites in magnetic field. The color denotes $|\langle E|\gamma_{x,s,\uparrow}|G\rangle|^2 + |\langle E|\gamma_{x,s,\downarrow}|G\rangle|^2$ in (g-i) and $|\langle E|\gamma_{y,s,\uparrow}|G\rangle|^2 + |\langle E|\gamma_{y,s,\downarrow}|G\rangle|^2$ in (j-l), where $s = 1, 2, 3$ is the site, $|G\rangle$ is the ground state, $|E\rangle$ is the first excited state. (g,j) At $\mu = 0$ the system develops edge states but there is no separated Majorana zero modes around the gap closing point. They are trivial bulk fermion states. As $\mu$ increases, two separated Majorana modes at site 1 and 3 appear near the gap closing point. Other parameters used: $U = 0$, $\Delta = 0.2$, $\alpha = 0.05$, $t = 0.05$, $T = 0.01$.

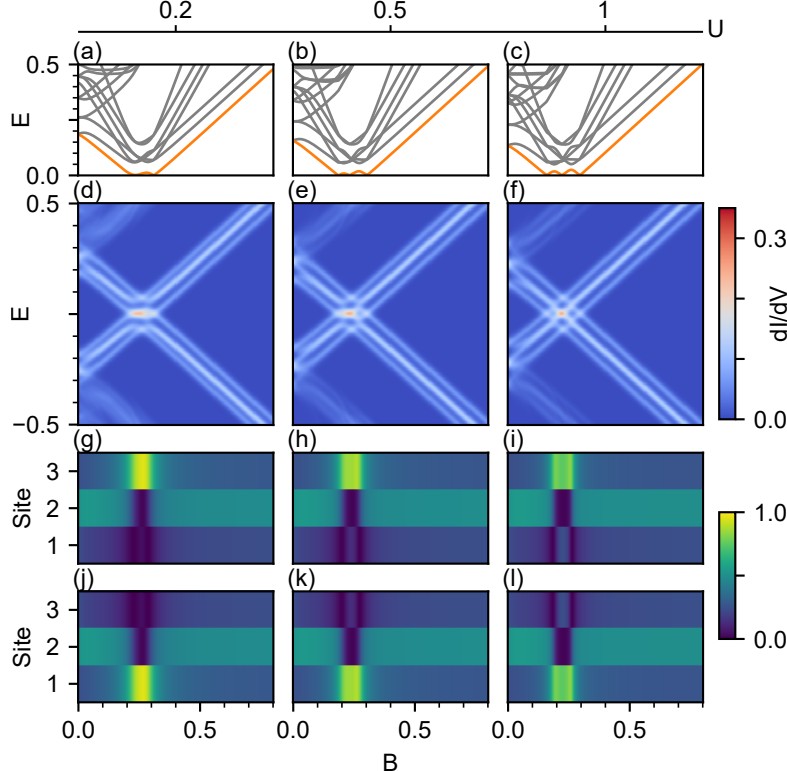

FIG. S2. Simulation results for a chain of 3 identical dots with the Coulomb interaction $U = 0.2, 0.5, 1$. (a-c) Many-body spectra versus the magnetic field. The orange line is the first excited state. (d-f) Differential conductance in magnetic field. (g-l) Evaluation of Majorana components (probability amplitudes) in different sites in magnetic field. The color denotes $|\langle E|\gamma_{x,s,\uparrow}|G\rangle|^2 + |\langle E|\gamma_{x,s,\downarrow}|G\rangle|^2$ in (g-i) and $|\langle E|\gamma_{y,s,\uparrow}|G\rangle|^2 + |\langle E|\gamma_{y,s,\downarrow}|G\rangle|^2$ in (j-l), where $s = 1, 2, 3$ is the site, $|G\rangle$ is the ground state, $|E\rangle$ is the first excited state. As $U$ increases, the extended zero-bias peak develops into three crossings and Majorana modes are no longer separated. Other parameters used: $\mu = 0.2$, $\Delta = 0.2$, $\alpha = 0.05$, $t = 0.05$, $T = 0.01$.

## DISCUSSION ON DIFFERENT MODELS

Refs. [18] and [19]: We were motivated by [18] and [19] in the early stages of the project. They propose to use a chain of quantum dots to realize an effective Kitaev model. However, significant details vary between these two proposals, and between the original Majorana proposals [5–7] and our experimental approach. Thus, our devices do not in fact implement or intend to implement the proposals in Refs. [18] and [19].

Model of Ref. [18] is for a linear array of quantum dots interspersed with superconducting islands. This model is an effective Kitaev chain when the nearest-neighbor hopping between quantum dots (through a superconducting island) is allowed and only a single spin-polarized level in each dot participates in transport. Spin-orbit coupling rotates spin polarization as an electron moves inside the dot, allowing proximity induced superconductivity that pairs electrons from neighboring dots rather than inside one dot (double occupation is forbidden in each dot). The model also requires crossed Andreev reflection which is a challenging experimental requirement for a chain of dots.

Model of Ref. [19] requires spinful quantum dots at single occupation. This model also features a tunable proximity effect between quantum dots and superconductors, which is achieved by applying a phase difference between two superconductors attached to a single dot. The tuning procedures described in Ref. [19] which include the tuning of chemical potentials, superconducting phases, and couplings are done at a fixed Zeeman energy, i.e., at a fixed magnetic field.

Our devices are geometrically and functionally different from those described above. However, they are also suitable for the realization of the Kitaev chain, as our own model, tailored to our 3-dot system, demonstrates.

Below we explain in what ways our devices are different from those proposed in Refs. [18] and [19]: In our devices, we have exactly three dots. Realizing chains of more dots as suggested in Refs. [18] and [19] with their geometry and the current fabrication techniques is challenging. Our device is not an array of alternating quantum dots and superconducting islands or a chain of quantum dots with superconducting loops, but a chain of Andreev quantum dots. In our device, each dot is strongly coupled to a superconducting lead, and has a singlet ground state (a hybrid of empty and doubly occupied states) at zero magnetic field. At finite magnetic field the ground states become spin-up Andreev Bound States. Dots have a quenched charging energy and exhibit no Coulomb blockade.

Devices characteristics captured by our model: Our model included in supplementary materials is not a literal representation of our device, but it captures key features of what we measure. Our model of three single level quantum dots is a generic description of a quantum dot chain with spin-orbit interaction, induced superconductivity and Zeeman effect. The spin-orbit length in InSb nanowires has been measured to be in the range of $100-200$nm [24, 25]. This is comparable to the distances between dots in our device. Spins undergo full or nearly full rotation while tunneling between the dots, over uncovered segments of the nanowire, where spin-orbit is better understood. This fact is captured in our model by setting parameters $\alpha$ (spin-orbit coupling) and $t$ (tunneling between dots) to be of similar values. Our model contains superconducting pairing for electrons from the same dot, not adjacent dots, which is consistent with the fact that we only observe Andreev bound states with singlet ground states in each dot at zero magnetic field. In the model, we use small values for $U$ (Coulomb interaction) to capture the lack of charging energy.

Models in Refs. [28] and [29] propose to couple several finite sized segments of topological superconductor through tunnel gates. The chain thus consists of multiple wire sections with individual superconducting islands. In each segment, Andreev bound states can be partially separated into Majorana modes. By coupling the neighboring segments, inner Majorana modes on each segment hybridize and leave the outermost Majorana modes unpaired, similar to a Kitaev chain. Models in Refs. [28] and [29] are relevant to our study since the wire segments are of a length similar to our devices. However, these models are too optimistic to assume partially separated Majorana already exist in each wire segment, which needs to be first experimentally established.

SUPPLEMENTARY FIGURES

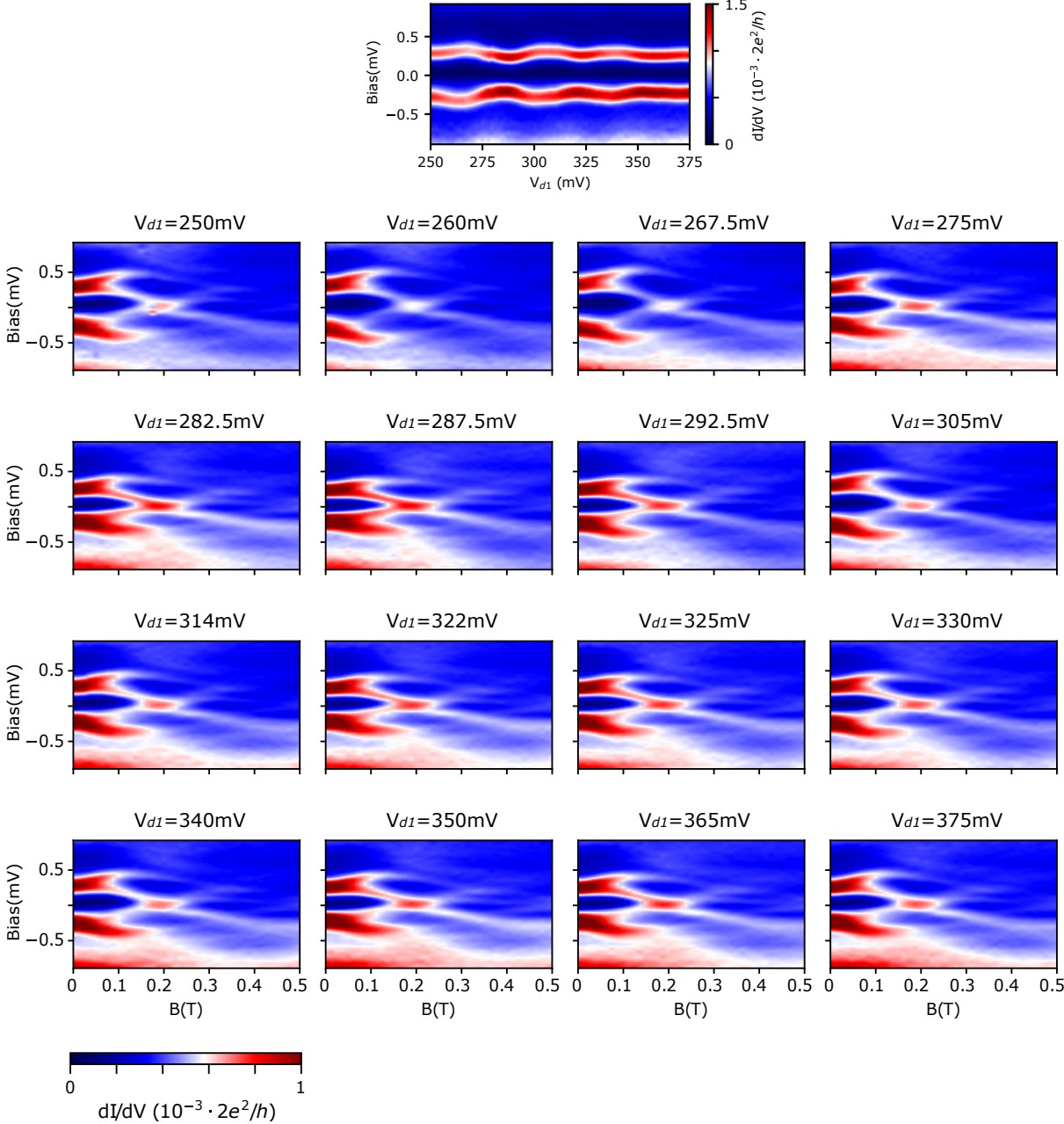

FIG. S3. Magnetic field evolution of the triple-dot Andreev bound states at different $D_1$ potentials. Gate voltage $V_{d1}$ is listed above each panel. For all panels, $V_{d2}$=576.5mV, $V_{d3}$=392.5mV. Measured from $N_1$ to $N_2$.

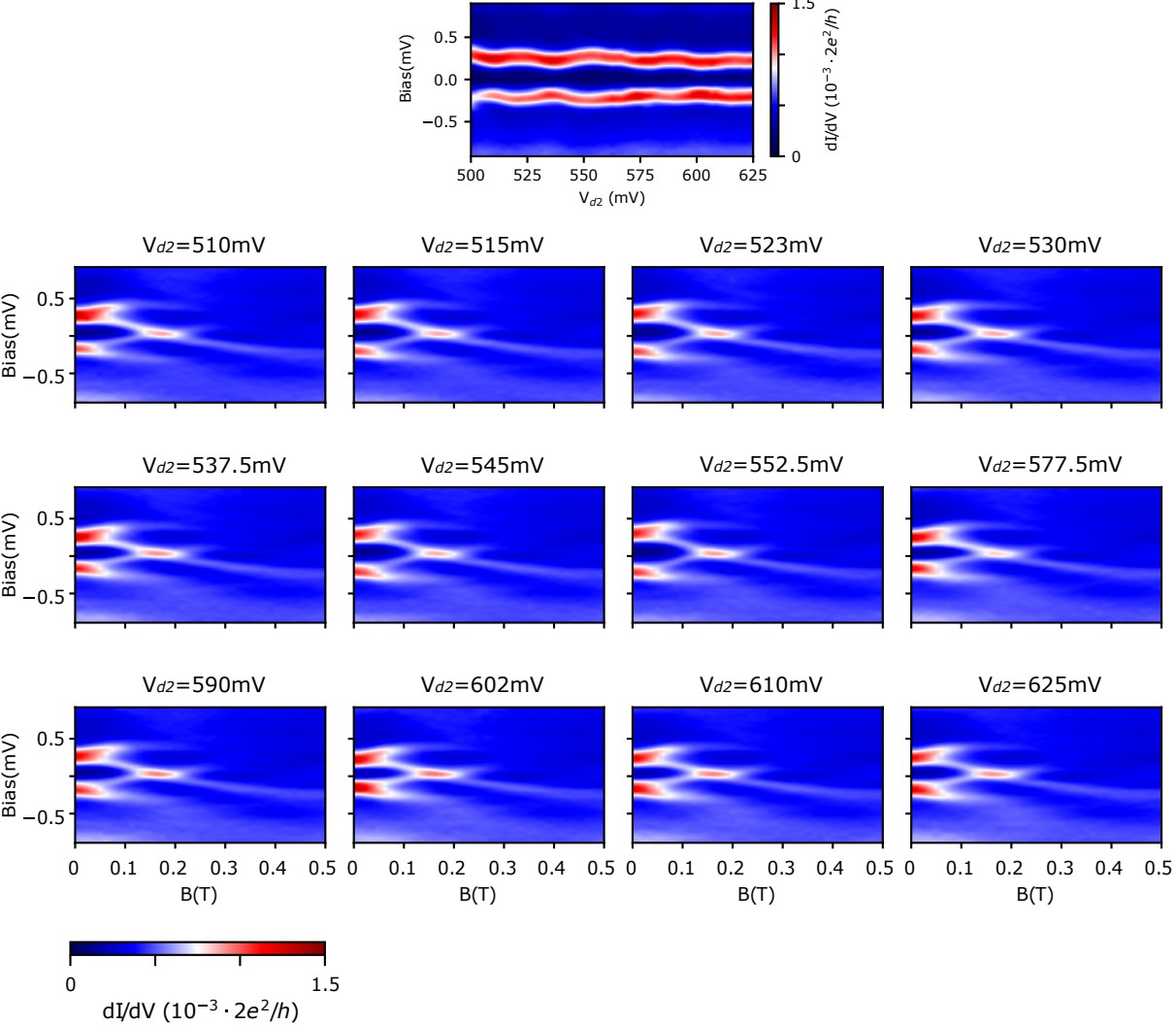

FIG. S4. Magnetic field evolution of the triple-dot Andreev bound states at different $D_2$ potentials. Gate voltage $V_{d2}$ is listed above each panel. For all panels, $V_{d1}$=322.5mV, $V_{d3}$=395mV. Measured from $N_1$ to $N_2$.

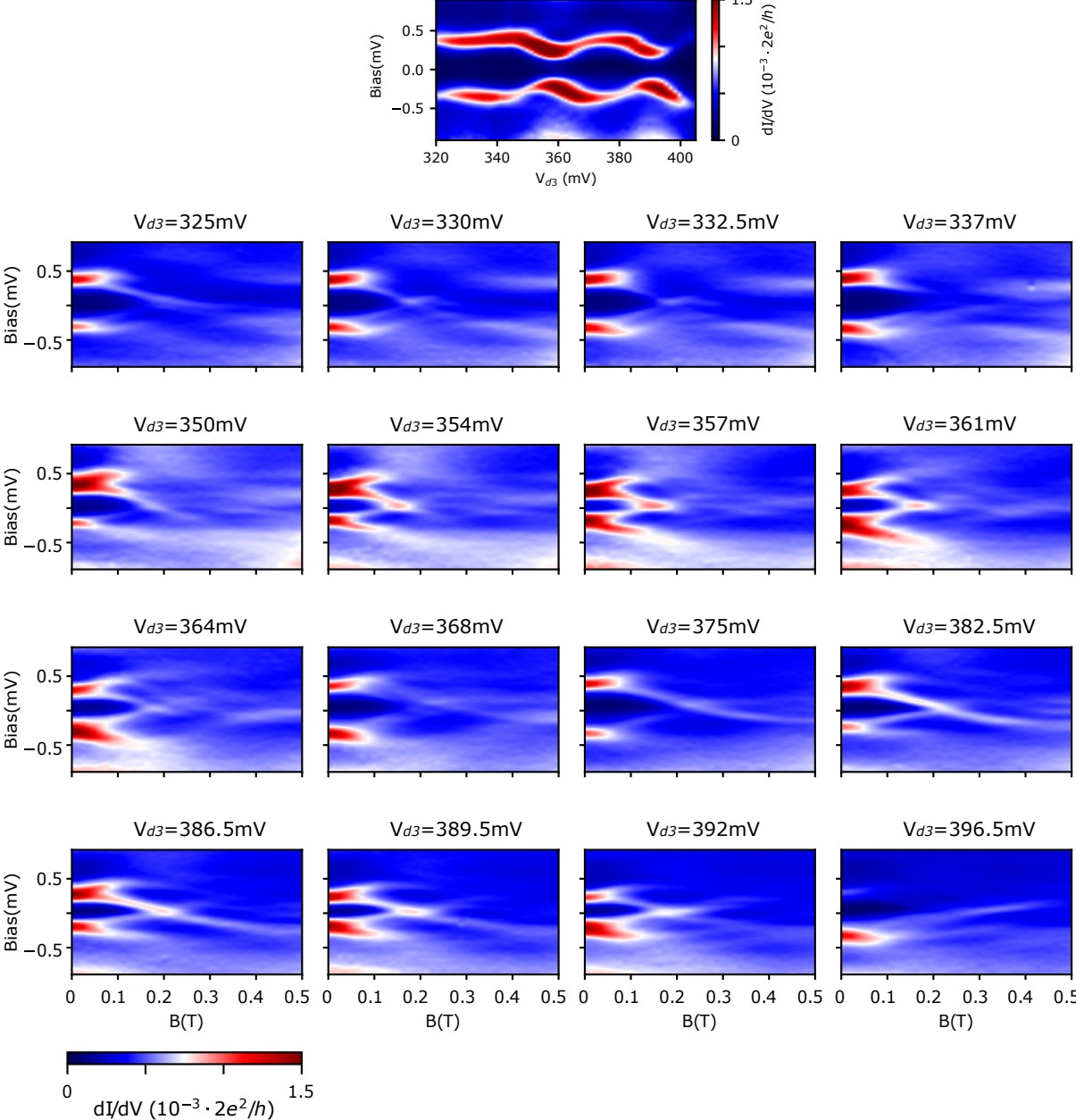

FIG. S5. Magnetic field evolution of the triple-dot Andreev bound states at different $D_3$ potentials. Gate voltage $V_{d3}$ is listed above each panel. For all panels, $V_{d1}=322.5$mV, $V_{d2}=576.5$mV. Measured from $N_1$ to $N_2$.

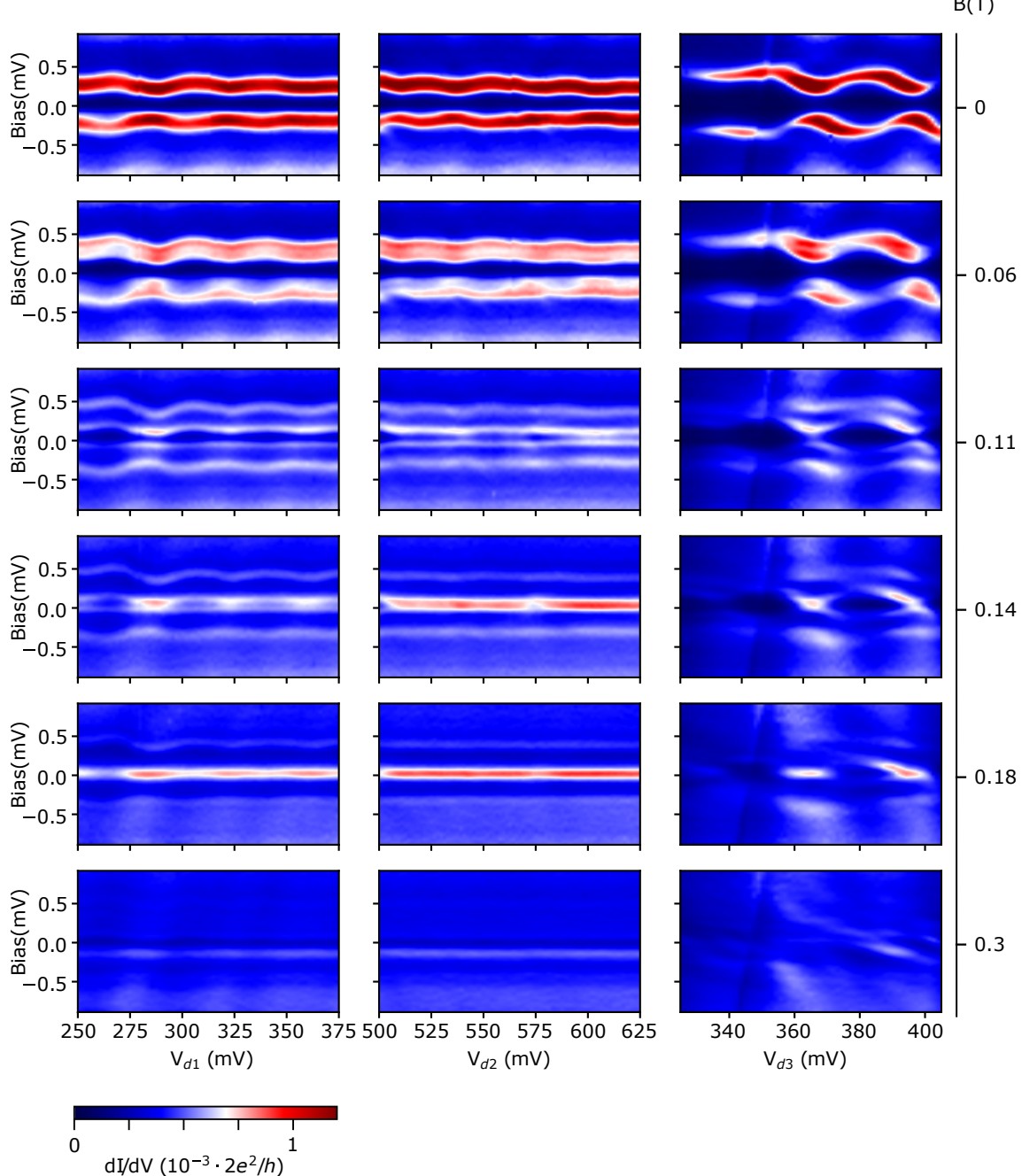

FIG. S6. Additional data to Fig. 2. Bias spectroscopy of Andreev bound states in each dot at different magnetic fields. Emergence of the first zero energy crossing. When tuning one dot, the other two gate voltages are fixed as: $V_{d1}$=322.5mV, $V_{d2}$=576.5mV, $V_{d3}$=395mV. Measured from $N_1$ to $N_2$.

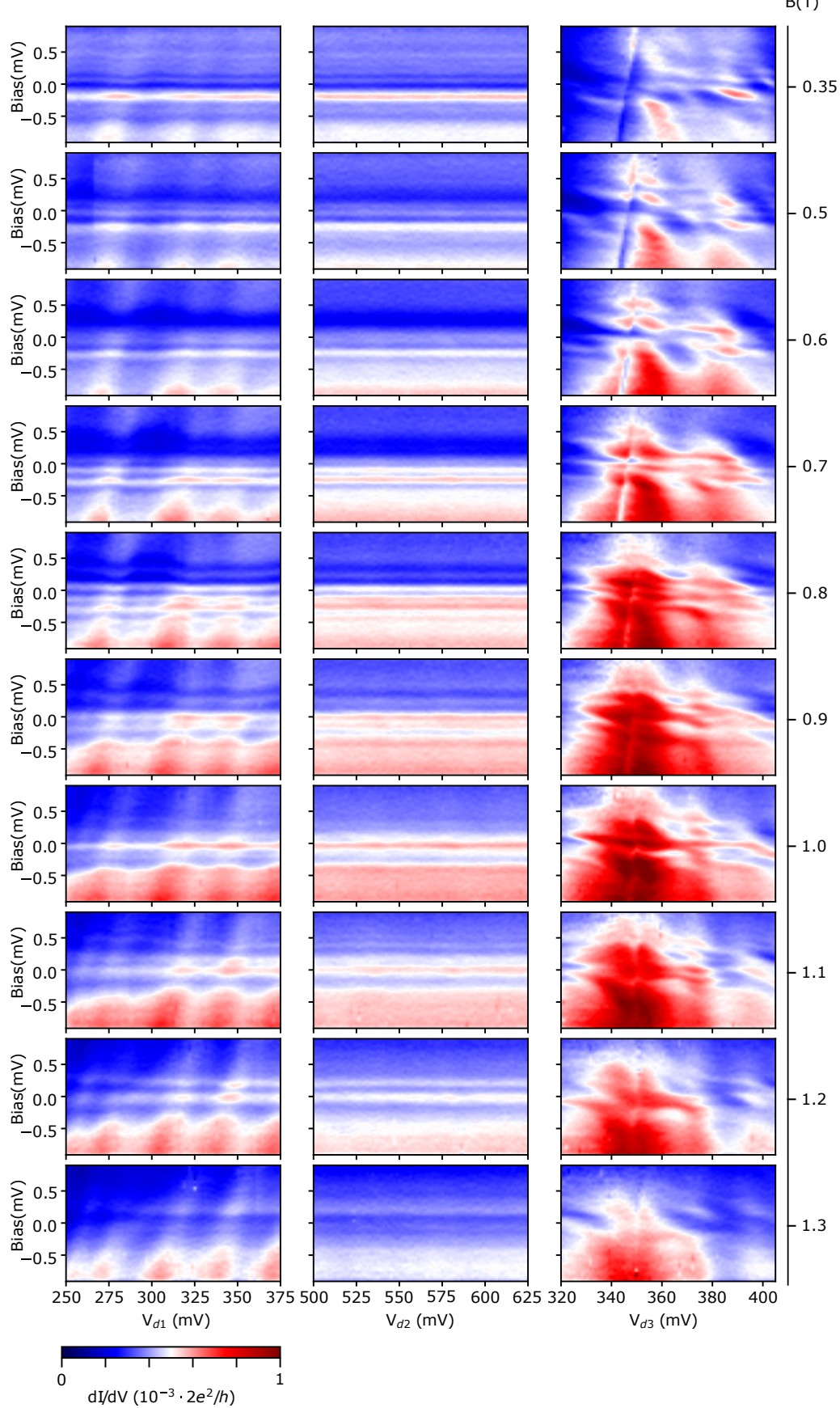

FIG. S7. Additional data to Fig. 2. Bias spectroscopy of Andreev bound states in each dot at higher magnetic fields. When tuning one dot, the other two gate voltages are fixed as: $V_{d1}$=322.5mV, $V_{d2}$=576.5mV, $V_{d3}$=395mV. Measured from $N_1$ to $N_2$.

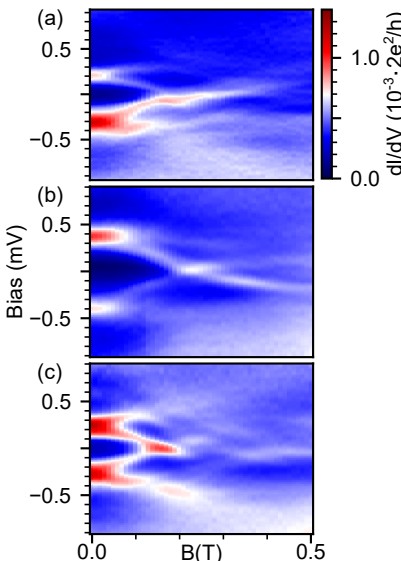

FIG. S8. Magnetic field evolution of the triple-dot Andreev bound states at different $V_{d3}$ indicated by short yellow lines in Fig. 4(c). (a) $V_{d3}=400$mV, (b) $V_{d3}=380$mV, (c) $V_{d3}=365$mV. $V_{d1}=185.0$mV, $V_{d2}=472.5$mV. Measured from $N_1$ to $N_2$. From Fig. 4(c), the onset point of the first zero energy crossing in magnetic field is $V_{d3}$ dependent. Fig. S8 provide more details on how the Andreev states evolve and how zero-bias peaks develop in magnetic fields. The spectra can be bias-asymmetric (a)-(b), or bias-symmetric (c). The first zero energy crossing can appear with different length in magnetic fields. We conclude the origin of the first zero energy crossing in magnetic field is Andreev states localized at $D_3$.

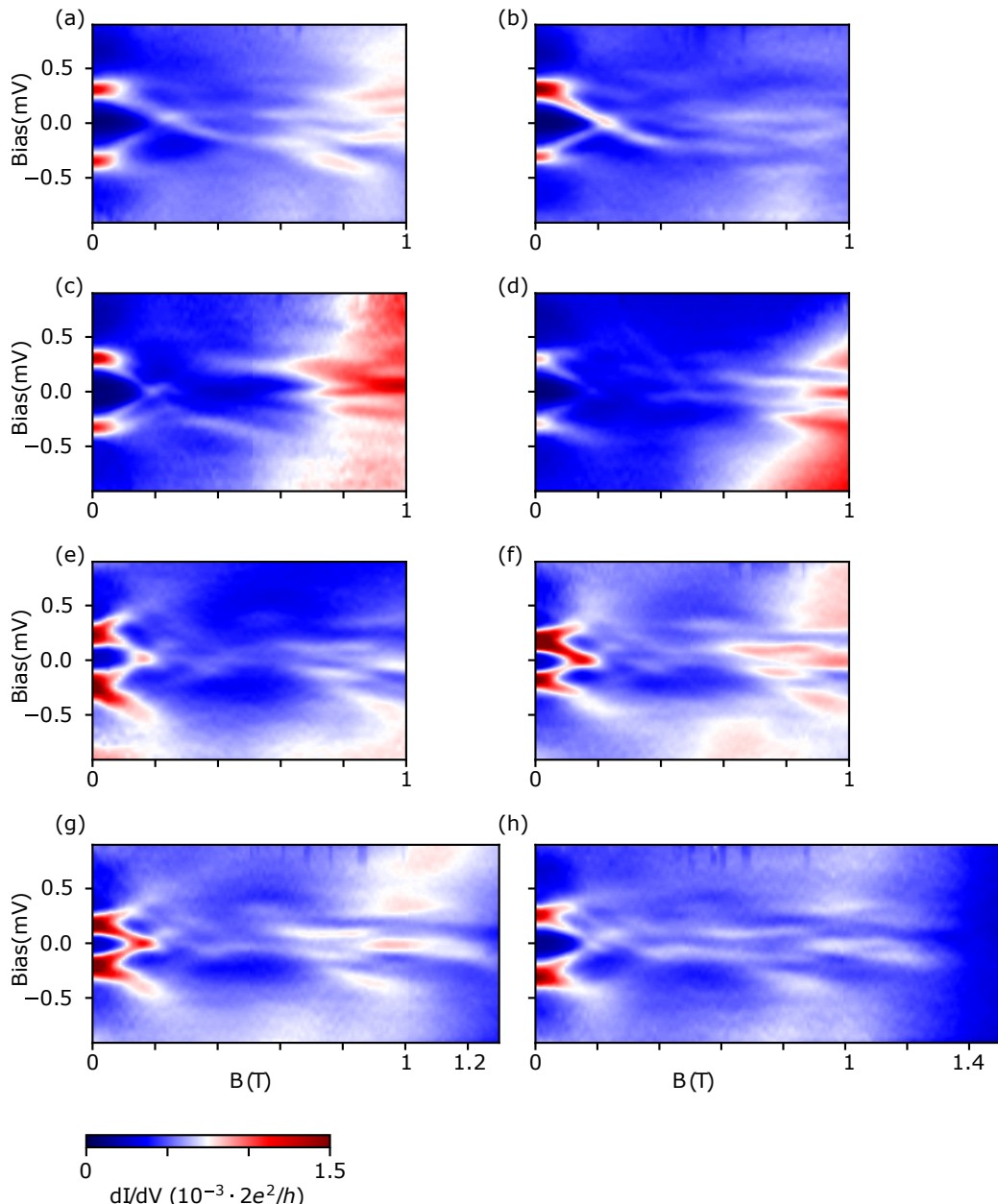

FIG. S9. Magnetic field evolution of the triple-dot Andreev bound states to higher magnetic fields, measured from $N_1$-$N_2$. (a) $V_{d1}$=320mV, $V_{d2}$=537.5mV, $V_{d3}$=377.5mV. (b) $V_{d1}$=288mV, $V_{d2}$=540mV, $V_{d3}$=390mV. (c) $V_{d1}$=320mV, $V_{d2}$=537.5mV, $V_{d3}$=335mV. (d) $V_{d1}$=285.5mV, $V_{d2}$=576.5mV, $V_{d3}$=370mV. (e) $V_{d1}$=322.5mV, $V_{d2}$=576.5mV, $V_{d3}$=360mV. (f) $V_{d1}$=320mV, $V_{d2}$=537.5mV, $V_{d3}$=362.5mV. (g) $V_{d1}$=320mV, $V_{d2}$=537.5mV, $V_{d3}$=365mV. (h) $V_{d1}$=288mV, $V_{d2}$=540mV, $V_{d3}$=375mV.

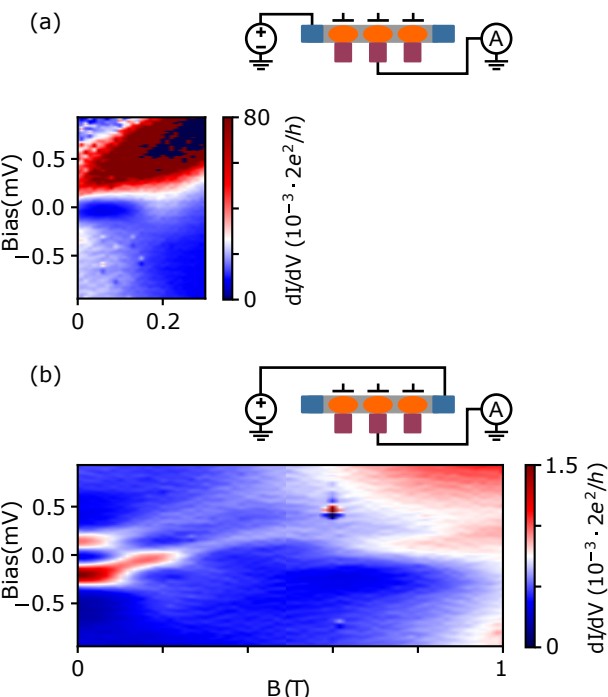

FIG. S10. Additional data to Fig. 5. Magnetic field evolution of the triple-dot Andreev bound states with different measurement configurations shown in circuit diagrams above panels, (a) $N_1$-$S_2$, (b) $N_2$-$S_2$. $V_{d1}$=315mV, $V_{d2}$=537.5mV, $V_{d3}$=396mV. Conductances are much higher in (a) than in (b), which indicates that the tunnel barrier located at the left end near $N_1$ is significantly more open compared to the right tunnel barrier near $N_2$. (b) is comparable to Figs. 5(a-b), suggesting the inter-dot barriers are also low. Thus whenever $N_2$ is excluded from the measurement configuration the resonances broaden considerably.

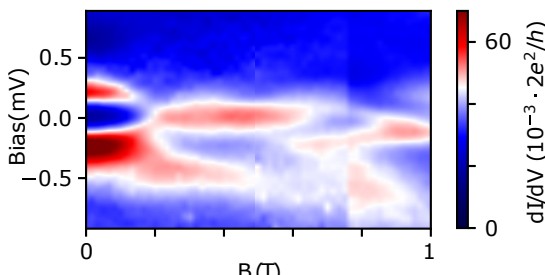

FIG. S11. Magnetic field evolution of the triple-dot Andreev bound states when inter-dot barriers are lifted, i.e. tune the quantum dot chain towards a continuous wire. Due to fewer and weaker tunnel barriers, conductance is much higher than measured with $N_1$-$N_2$ configuration, and the resonances are broadened. Instead of clear splitting of Andreev states, zero energy crossing, and resonances at low voltage bias beyond the first crossing shown in Fig. S9, a long zero-bias peak extends for near a half Tesla is present. The differences might be related to increased inter-dot couplings as lowering inter-dot barriers. However, a conclusion cannot be drawn based on these broadened resonances.

## DATA FROM ANOTHER DEVICE

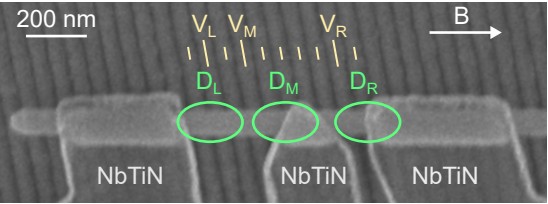

FIG. S12. SEM of device B with a different triple-dot design. In an InSb nanowire, three dots are defined to the side of superconducting contacts by fine gates (indicated by yellow lines). Quantum dots defined this way have weaker but gate-tunable coupling to superconductors compared to the triple-dot design used in the main text. ($V_L$, $V_M$, $V_R$) are plunger gate voltages tuning potentials in the left, middle, and right dots ($D_L$, $D_M$, $D_R$). All 3 electrodes are superconducting NbTiN. The middle lead has a small contact area to reduce stress. Transport measurements are performed using the two outer leads as the source and drain while floating the center lead. Lack of non-superconducting leads may result in additional resonances in spectra and complicate the interpretation of the triple-dot states [70].

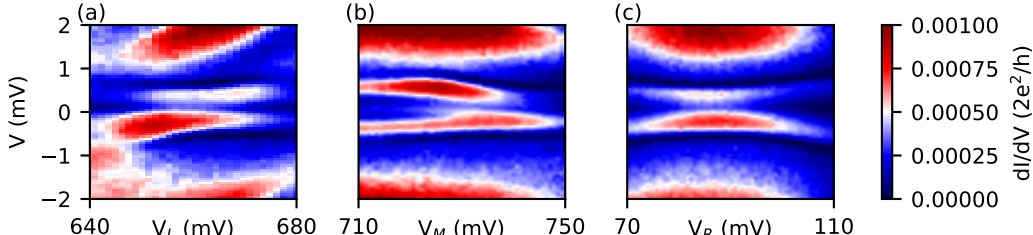

FIG. S13. Bias spectroscopy of (a) $D_L$, (b) $D_M$, (c) $D_R$, B=0. Anti-crossing resonances indicate Andreev bound states in all three dots have singlet ground state.

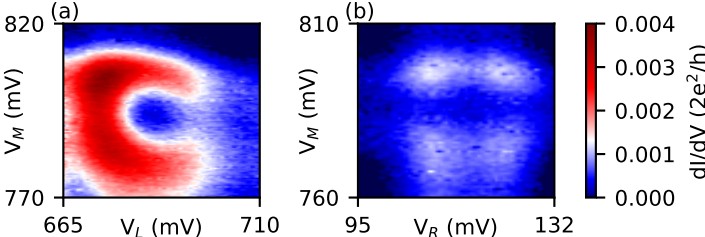

FIG. S14. Stability diagrams of dot pairs (a) $D_L$-$D_M$, at 0.25 mV, and (b) $D_M$-$D_R$, at 0.3 mV. B=0.

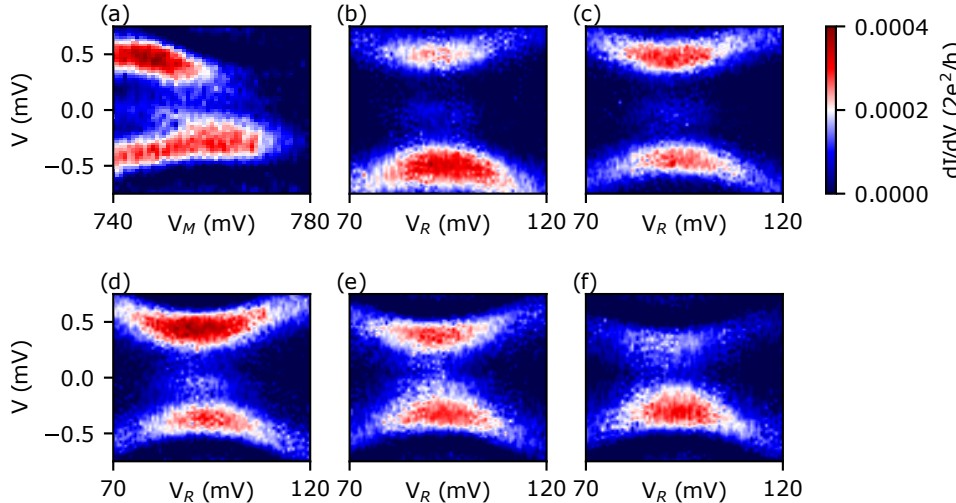

FIG. S15. Bias spectroscopy of $D_R$ with different $V_M$, at B=0. (a) dI/dV vs. $V_M$ as a reference for $V_M$ settings in the rest panels. (b-f) $V_M = 730, 740, 750, 755, 760$mV.

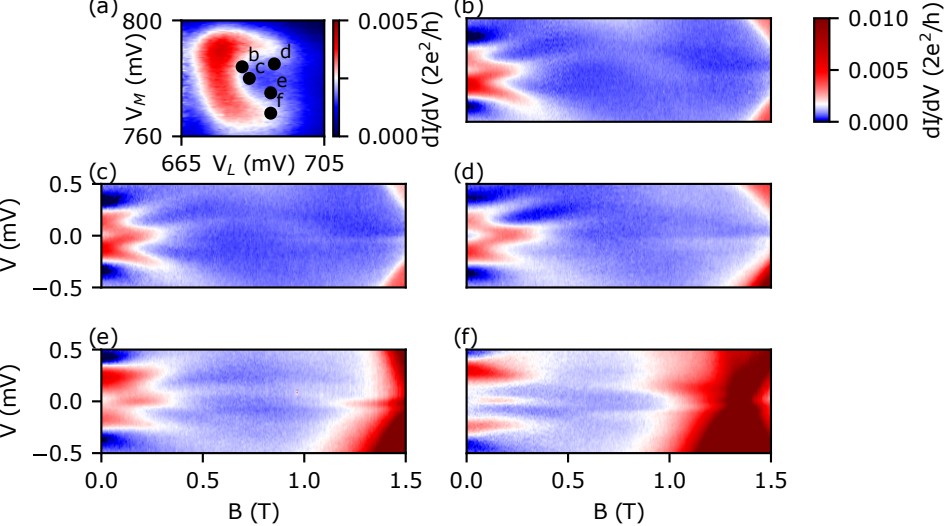

FIG. S16. Magnetic field evolution of the triple-dot Andreev bound states at different gate configurations. (a) Stability diagram of $V_L$-$V_M$, as a map for gate configurations used in the rest panels. $V_R$ is kept the same during the measurement. (b-d) The evolution resembles Zeeman splitting of trivial Andreev bound states. (e-f) A short and extensive zero-bias peak appears.

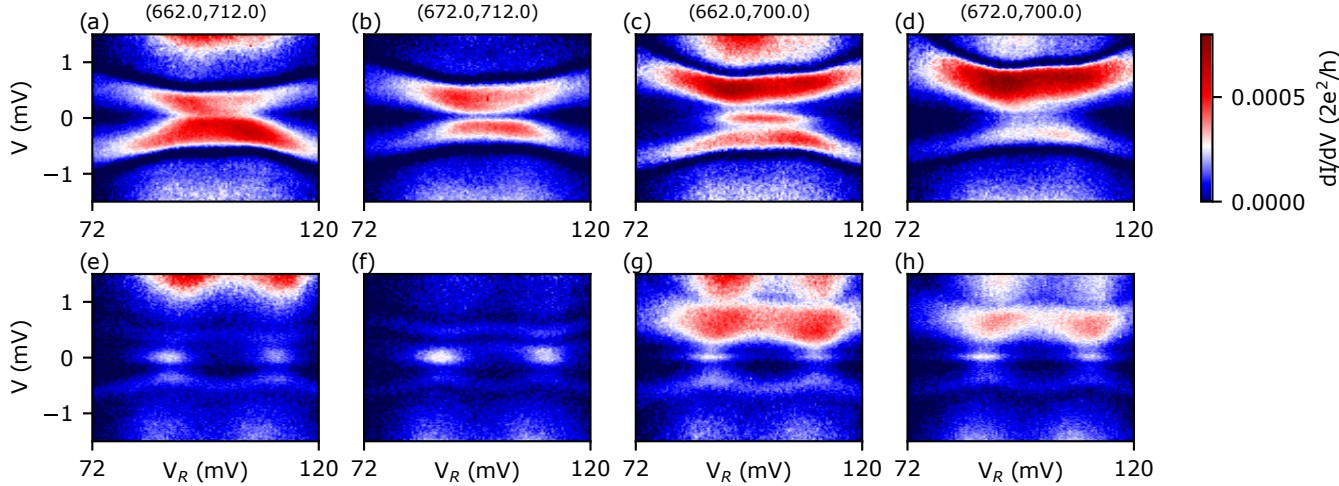

FIG. S17. Bias spectroscopy of $D_R$ with different $(V_L, V_M)$, at (a-d) B=0 and (e-h) B=0.2T. Gate settings of $(V_L, V_M)$ in mV are listed on top of each column. From B=0 to B=0.2T, ground state transition from singlet to doublet [20].