# Peer review of "Triple Andreev dot chains in semiconductor nanowires"

_SciPost Physics_

## Round 2 · Referee Report · Elsa Prada (Referee 1) · 2021-10-2

Report
In this work the authors bring into practice the so-called Kitaev chain model, or at least one version of it. For that they use an InSb semiconducting nanowire and with the help of several back gates, they confine three quantum dots (QDs) in series, attached to source and drain contacts. Moreover, these dots are coupled to superconducting NbTiN contacts (in the shape of "fingers" deposited above the QD regions), bringing in principle these dots into the superconducting state. In this way, they create a chain of three "Andreev" QDs. Subsequently, they perform transport measurements of current and differential conductance. Particularly, they apply a voltage bias through the left contact, and measure current and differential conductance from the right contact while floating the superconducting fingers. By changing parameters such as the back-gate voltages below each dot region and an applied axial magnetic field, they look for signatures of Andreev levels inside the superconducting gap and analyze their behavior. The goal is to find zero bias anomalies compatible with the existence of non-local Majorana states at the outer dots of the device.
Let me start with some general comments and then I will get more specific:
I find the subject of this work is of high interest for the community of hybrid semiconducting nanowires, Andreev quantum dots and for the search of Majorana states in this kind of platform. Having analyzed in depth in the literature the "continuous" nanowire model (although this is still an ongoing effort), it is only natural to try to start studying experimentally more sophisticated models such as this Kitaev chain. If I understand correctly, this is one of the first (if not the first) attempts to try to bring this model into experimental reality. For that alone, I find this study is worth considering and publishing.
To approximate a chain just out of three sites (three dots) looks a bit wishful, at least from the point of view of a theorist, but this is definitively a first step. Besides, we all must recognize the tremendous ability and craftsmanship to get a working device of this type must have taken. As the authors note: "Even though the desired regime is not accessible, this device with new geometry has over- come many nanofabrication challenges...". "These experiments took several years to complete. Device fabrication is challenging and required significant development to realize multiple narrow superconducting contacts to a nanowire. Measurements are time-consuming due to the large number of experimental parameters (7 gates and 5 contacts). Yes, in the future it would be interesting to use different superconductors or different fabrication methods such as SAG, but this cannot be switched out and tried without a huge investment." I can only praise the authors for this effort, and again, that itself deserves publication in my opinion.
Another issue is what is going on exactly inside such a complicated device and whether this type of system is better or presents practical advantages to create Majorana bound states than the continuous wire one. With respect to the first question, the authors carry out an extremely simplified model (I would even say naive), but for me that is enough because this is essentially an experimental paper and it is our job as theorists to make sense of the experimental evidence. In any case, I have several observations about this below. With respect to the second question, I honestly think that if it is difficult to get the right conditions for Majorana state formation in the continuous nanowire device, it is only going to be worse in this type of multi-section wires. More on this later.
This said, I still find that this line of research is worth pursuing, since advantages may appear or be possible in the future, and definitively it will bring to the community a better understanding of these wires and Andreev states inside them. I see this work as a first attempt to build an Andreev Kitaev chain. The results are negative, meaning that the necessary conditions for the topological phase are not met, but they are presented with honesty and many interesting conclusions can be driven out of them. For example, the necessity to have rather similar or homogeneous QD's and good proximity effect in all of them. Thus, this work can be seen as a “towards a simulated Kitaev chain” paper, as another Referee put it. This is perfectly fine for me, we cannot pretend that one single study, particularly as difficult as this one, solves all the challenges at the first try. Surely many more studies, both experimental and theoretical, will be necessary to bring this idea to success. This in itself is also good because it can stimulate discussion and bring citations to the journal.
Finally, I think this paper is very well written. I also appreciate the clarity and honesty with which the results, specially the negative ones, are presented, and the effort to bring the non-expert audience to the discussion with sections such as "limitations", "further reading", "background" and "discussion of different models".
Now I proceed with several comments and criticisms:
1) In the Introduction, and as a motivation for their work, the authors (only) cite Refs. [18] and [19]. However, as the PRL Referee noted and the authors agreed, this work is not trying to emulate those types of chains precisely. The authors themselves added a very useful section "Discussion on different models" where it is clear that the type of device they build is more similar to Refs. [28,29]. I am more familiar with [29] and, indeed, I find that this work is similar or perhaps identical in spirit to Ref. [29], and has less to do at the end with [18] and [19]. I think this fact should be acknowledged already at the level of the Introduction to avoid misinterpretations or confusion. Concerning citations, I want to bring another reference to the attention of the authors, because I think it is pretty much related: "Effects of the electrostatic environment on superlattice Majorana nanowires", Phys. Rev. B 100, 045301 (2019). In this work there is only one back gate below the semiconductor, but otherwise it is very similar, with several superconducting fingers on top of the wire. Among other things, this superlattice creates and inhomogeneous potential along the wire in the form of potential barriers and wells similar to what it is studied in Ref. [29] and what must be happening in the author's device. In this work we study carefully all the parameter variations produced in this complicated device when taking into account the 3D geometry and electrostatic environment realistically, and their effect on the phase diagram and Majorana formation. More on this below.
2) In the Introduction the authors rightly mention: "Theory suggests a robust topological superconducting phase in chains of Andreev quantum dots [18, 19]." This is a positive message worth noting. However, while this might be true for the proposals of [18,19], I want to mention that the phase diagram of a superlattice of superconducting fingers is in general more complicated and less robust that that of a continuous wire model. As can be seen in Phys. Rev. B 100, 045301 (2019), there are several types of phenomena that break the topological phase and that are summarized in Fig. 10 (the formation of localized sates, longitudinal subband gaps and longitudinal subband overlaps.). These or similar weaknesses (with a different language) are also found in Ref. [29]. Since the device the authors study is actually similar to these works, this could be at least mentioned.
3) The authors use a many-body description for their Andreev states (singlet ground state, doublet excited states, many-body spectra, etc.). However, they themselves acknowledge that electron interactions are negligible in their experiments: "Dots have a quenched charging energy and exhibit no Coulomb blockade.", "The charging energy is quenched because the dots are covered by huge superconducting leads. It can be assumed to be zero. There are no Coulomb diamonds present in the system." Thus, for me, it doesn't make much sense to use such a many-body interpretation, it is confusing. In my opinion, the authors measure single-particle resonances, Andreev levels, that split under magnetic field.
4) Concerning the previous point, the authors perform a theoretical (simplified) study in the supplementary material. There they introduce interactions, U. If they believe there are no interactions in their device, it is perhaps a bit strange to study precisely the effect of U. Perhaps they could motivate such study a little. Actually, I find it interesting since with this simplified model one can already learn that interactions are detrimental for the topological phase. By the way, the authors should state how they compute the effect of interactions. I assume they use a self-consistent Hartree approximation.
5) I must say that the theoretical model section is rather short and contains very few explanations. I had to make a considerable effort to understand what the authors were trying to prove and the conclusions they were extracting from their simulations (these were only mentioned, not explained, in the figure captions). For example, they could explain a bit more why they calculate the Majorana probability and what's its meaning. Moreover, with just a couple of figures of differential conductance and Majorana probability amplitudes for some parameters, the authors imply that they have their device figured out. See for instance statements such as "While we are motivated by early theoretical proposals, we do not aim to replicate them and instead develop a theory tailored to our devices (see supplementary information for discussion).", "... For such a short chain, within a narrow parameter window, the probability distribution of Majorana wavefunctions indicates a partial separation of two Majorana zero modes localized at two end sites (see supplementary materials for simulation results)." As I said at the beginning, I think this is an experimental paper and as such, I consider it is more than enough. I even appreciate the effort the authors do to analyze a simple toy model to get an idea of the physics behind the experiments. However, I don't really think the authors perform any kind of serious study of the topological phase or the real behavior of such a sophisticated device, and this should be acknowledged somehow. The authors should refer there to more serious works such as Ref. [29] or Phys. Rev. B 100, 045301 (2019).
6) Related to the previous point, at the end of Section "Discussion of different models" the authors say: "Models in Refs. [28] and [29] are relevant to our study since the wire segments are of a length similar to our devices. However, these models are too optimistic to assume partially separated Majorana already exist in each wire segment, which needs to be first experimentally established." I don't think this is correct. These works (as well as Phys. Rev. B 100, 045301 (2019)) do not "assume" partially separated Majoranas. This partial separation happens immediately for wire segments of the order of the QD regions of their device simply becase these wires have spin-orbit interactions and magnetic fields. The wavefunction in their device is surely not a combination of three differentiated almost point-like wavefunctions in each of the QD regions weakly coupled through tunneling barriers. As the authors themselves acknowledge, the barriers between QD regions are low (or even absent), and thus the wave function is basically spread along the whole wire. When they observe zero bias anomalies, most probably they have quantum well regions between the covered regions by the superconductors, see for instance Fig. 1(c) in Phys. Rev. B 100, 045301 (2019), where the realistic electrostatic problem has been solved. Thus, most probably the shape of the wave function, in the best case scenario, is similar to Fig. 12(h) in Phys. Rev. B 100, 045301 (2019) (but with fewer superconducting fingers) or to Fig. 9 in Ref. [29].
7) Finally, let me mention that the Majorana localization length for a periodic structure is larger than that for a continuous model wire, see Phys. Rev. B 100, 045301 (2019). Thus, for a similar nanowire length, the Majorana wavefunction overlap is typically larger and the topological mini gap smaller.

Anonymous on 2021-09-24 [id 1785]
Editor's remark: the referee report from Physical Review Letters has been posted here by Professor Frolov with the permission of the PRL editors.
Sergey Frolov on 2021-09-24 [id 1784]
Here is our (authors) response to a PRL referee. We thank the referee and welcome the referee to continue this discussion!
RESPONSE: Our study is inspired by the theoretical proposals Refs. [18] and [19]. However, it is not our goal to realize these proposals. We didn’t design our experiment according to all details from Ref. [18] or [19]. We also build our own model to better describe our devices (in supplementary).
In v2 of the manuscript we have added a discussion in supplementary to compare the various models and their assumptions.
RESPONSE: Spinful QDs with single (odd) electron occupation have doublet ground states. From the Andreev resonances and their magnetic field evolution, the QDs in our device only have singlet ground states at zero field. However at finite magnetic field where we see zero-bias crossings of energy levels, the dot(s) undergo(es) a quantum phase transition into a phase with a spinful ground state. We are capable of tuning through several occupations with our gates.
RESPONSE: The metalization problem and heavily reduced spin-orbit strength is more of an issue for aluminium, where the proximity induced gap is nearly the size of the bulk gap. However, for NbTiN, the decrease of spin-orbit interaction is less severe, as the induced gap is 5-10 times smaller than the bulk gap. And NbTiN has a larger spin-orbit interaction than Al.
Even though the spin-orbit length in the nanowire sections under NbTiN superconducting leads can be shorter than in bare nanowire, the distances between dots (nanowire sections NOT covered by superconductors) are comparable to the spin-orbit length (100-200nm). Hence substantial, if not complete, spin rotation can be achieved by tunneling between the dots.
REPONSE: Soft gap does not prevent the spectroscopy of Andreev states in quantum dots, as has been demonstrated on many occasions. The soft gap only poses as challenge for the realization of topological quantum bits, where any subgap density of states is a decoherence mechanism.
These experiments took several years to complete. Device fabrication is challenging and required significant development to realize multiple narrow superconducting contacts to a nanowire. Measurements are time-consuming due to the large number of experimental parameters (7 gates and 5 contacts). Yes, in the future it would be interesting to use different superconductors or different fabrication methods such as SAG, but this cannot be switched out and tried without a huge investment.
For example, ‘epitaxial’ materials come without leads and possess large charging energy. Grounding those islands is a tricky process, which may be possible with newly developed SAG wires that appeared after this project was already well underway.
RESPONSE: Phase coherence is not necessary to obtain an effective Kitaev chain. Our own model does not include that control knob.
RESPONSE: we only can talk about what is visible in the data. We see a wiggly ABS-like resonance that is present around 0.5 mV . We do not have a ‘pseudogapmeter’.
RESPONSE: The charging energy is quenched because the dots are covered by huge superconducting leads. It can be assumed to be zero. There are no Coulomb diamonds present in the system.
RESPONSE: we do not compare our results to Ref. 19. Also, that paper, as all other relevant papers, provide a qualitative guide at best as they are way too simple to describe an experimental system.
RESPONSE. In our model, every dot is single-orbit. Energies are in the unit of eV. The referee is correct that any match to a model is qualitative. In particular the purpose of our model is to demonstrate that Majorana separation in a three-dot chain, if achieved, is fairly intermittent for the rough parameters of the system such as the ratio of spin-orbit energy and the apparent gap.
RESPONSE: we have share full data from this experiment. Including a 150-slide powerpoint file with pictures of data, on Zenodo. All barriers were tuned and the dots are definitively under the superconducting leads.
RESPONSE: We can no longer do the exact measurements requested because the device has been warmed up with COVID hit in March 2020. But we have obtained extensive data to prove this point.
RESPONSE: this is why we worked out our model. We were hoping to see states from all three dots hit zero bias at the same magnetic field. This would have created the conditions for the Kitaev chain. It would still not have proven Majorana separation. Then, probing the middle dot we would expect to see no ZBP, while at the same time ZBPs on the left and right dots. The peaks can indeed be somewhat extended in magnetic field, in particular what the crossing points are somewhat spreadout.
RESPONSE: there is no Coulomb blockade here.
RESPONSE: all that can be measured was measured. As we commented above. Some of these parameters cannot be known fundamentally by this or another measurement technique, such as spin-orbit under a metal.
RESPONSE: THe referee came up with these bullet points based on a theory paper, but there is not a well established bullet point list for how to go about demonstrating a Kitaev chain. Nor were we trying to implement models in particular papers that the referee read to create their bullet list.
RESPONSE: clarity can always be improved. We hope that v2 of the manuscript is more clear.
RESPONSE: We encourage the referee to find a more sober experimental paper in this field. Would be interesting to read!
RESPONSE: The referee likely means Fig 2. For hard induced gap, E_1+E_2+E_3 is necessary for transport through 3 dots in sequence. However, for soft gap, tunneling could happen at lower voltage bias. This is a clear advantage of the soft gap because it makes the spectra of all dots visible in parallel.
RESPONSE: We love this colorscale, we find it clear and have used it in numerous published papers.
RESPONSE: we will improve this. Thanks!
Sergey Frolov on 2021-09-24 [id 1783]
This Report has been obtained to a previous version of the paper, 2105.08636v1, when it was under review at PRL. The authors have already addressed these comments in 2105.08636v2, along with comments by other referees that were minor. The authors will post a response to this Report as a separate comment.
The authors present an experimental study of transport through a device consisting of 3 quantum dots (QDs) in series which are each coupled to a superconducting lead. The authors claim that theirs is in principle a suitable device for the simulation of the Kitaev chain and thus for the observation of Majorana end modes under suitable tuning conditions. As they mention in their paper, they do not reach these conditions with their device as the Andreev bound states (ABS) in one of the QD-superconductor interfaces does not reach zero energy at the same narrow magnetic field window as the other two QDs.
The authors’ narrative entails that they could have reached the necessary conditions if this unfortunate detail would have worked to their favor. Regrettably, the authors omit other ingredients which are necessary to simulate the Kitaev chain as per Refs. [18] and [19] in their paper. Since their device is closer in nature to the proposal in Ref. [19], to this end I list the missing ingredients indicated there below:
-The QDs must be spinful, that is, all tuned into single electron occupation. In the device of the authors, the occupation of the QDs is unclear.
-The strength of the spin-orbit (SO) coupling was not measured, so it is not known if even in principle the QDs below the superconductors can realize the Kitaev model. It is not enough to quote bare InSb nanowire SO strength, given that it is expected to be decreased (by some amount and can even become negligible) when the semiconductor is in contact with the superconductor. This is known as the metallization problem. Due to this, it is also not known how the QD size compares to the SO length. The direction of SO is not measured either, so it is not known which direction of applied magnetic field is needed. There are known methods which can be used to measure SO strength and direction in QDs.
-The superconducting gap is not hard, and therefore the authors cannot speak of “subgap” ABS in their device since it does not have a true gap. Incidentally, they would not have been able to speak of subgap Majorana end modes in their device if they had reached the ideal conditions, either (as a soft “gap” is not a gap). Why wasn’t the superior and now routine in-situ grown epitaxial superconductor/semiconductor technology used to obtain a nanowires with a hard gap and thus true subgap states?
-Each superconductor induces a different superconducting phase in the QDs. Without controlling the phase in each superconductor through superconducting loops (not present in their device), how do the authors make sure that the phase-matching condition for the Kitaev chain is reached?
-Nearly all the important experimental parameters of the authors’ device are unknown. Besides the ones already mentioned, these are: A. What is the size \Delta of the “superconducting” pseudogap? B. Coulomb blockade seems detrimental to the localization of Majorana end modes according to the author’s model. What is the Coulomb energy? Since the Coulomb energy is unknown, I do not know if I should think of the QDs as non-interacting or interacting levels, and I do not know how to interpret subgap colormaps in Fig. 2 and stability diagrams in Fig. 3. Plots showing Coulomb diamonds for each QD are needed. C. The single-level approximation is used in Ref. [19]. What is the level spacing in their device? How does it compare to other parameters to make sure that the single-level approximation is valid for their case?
-Importantly, since the parameters of the system are not measured, their model is not a reflection of the device. In their model, the authors show in Fig. S1 plots at various \mu values without indicating their relation to the device. Does \mu=0 correspond to zero electrons in the QDs? What about \mu=0.2? How to interpret the values of U in Fig. S2 if there are no units given for this nor any of the other model parameters? If units cannot be quoted, at least ratios should be.
-The authors do not conclusively show that the QDs are located below the superconducting leads. To this end, they should provide measurements like those in Fig. 3 as a function of the gates which tune the bare segments of the InSb nanowire. If conductance lines in these other colormaps have a weaker dependence on these other gates, indicating a weaker lever-arm parameter (and if no other QD lines are seen), then that would constitute better evidence of the authors’ claim. An alternative is to make measurements as the ones in pages 81-84 of the PhD thesis of Gramich (https://edoc.unibas.ch/43462/). These measurements demonstrate that a QD lies underneath the superconducting contact, while the authors’ measurements do not conclusively do so. The authors’ device is a more complicated version of the carbon nanotube device in Fig. 5.4 in that PhD thesis, but the same methodology should apply.
If the authors had succeeded in their approach, could they please sketch what kind of features would they have expected to observe in the data in Fig. 1? Continuous (in gate voltage) zero-bias peaks (ZBPs) in all colormaps, or discontinuous (in gate voltage) ZBPs? And if discontinuous, for which occupation of the QDs? Would the ZBPs be present in Coulomb blockade or in the charge-fluctuation regime? Same question for Fig. 4. As a more general question: Which would be the signatures of Majorana end modes in their type of device? Compounded by the lack of measured device parameters indicated above (Coulomb charging energy, Delta, level spacing, SO strength, SO length, SO direction), the lack of this kind of information will prevent someone from reproducing their experiment or from bringing it to its final goal.
In summary, while the paper by the authors is motivated as a “towards a simulated Kitaev chain” paper, it is not rigorous enough nor it meets enough bullet points in the criteria to be put into that category. It is also hard to follow due to lack of sketches/explanations of what is expected if their device would be the ideal realization of the Kitaev chain versus what the authors think it is happening in their device. I believe that if the paper’s claims were sobered and written in a way such that they were rigorously and clearly supported by the data, then the paper could make a match for a more specialized Physical Review journal such as PRR or PRB (after major revisions).
Minor comments below:
How should I understand the convolution of ABS in Fig. 1? Does that mean that the excitation energies of the 3 ABS add up? E_1+E_2+E_3? Or does that mean that E_1=E_2=E_3 is needed to observe something in the spectrum? Are the ABS hybridized forming bonding-antibonding states due to the inter-QD tunnel coupling? A sketch would help.
I recommend the authors to change the blue-white-red colorscale to a linear colorscale to avoid the abrupt changes in color which occur when the white color is reached, and to let the reader appreciate better the softness of the pseudogap. One plot which shows the pseudogap in log scale at 0 field and at finite field would be helpful, too. Alternatively, the authors could fit their pseudogap with a Dynes density of states and quote the Dynes broadening parameter.
The supplementary data in Figs. S3 to S17 would be helpful to the reader if the caption or an accompanying text described what is happening and what can be learned from the colormaps. The amount of data is overwhelming as the features are not properly described.

---

## Round 2 · Referee Report · Falko Pientka (Referee 2) · 2021-11-14

Report
The paper concludes with a thorough discussion of experimental challenges and limitations of the current device geometry.
I certainly appreciate the endeavor to build superconducting quantum-dot arrays, which may be interesting even beyond the main motivation of this paper - building a Kitaev chain. I also think the discussion of the advantages and drawbacks of this particular geometry, e.g., in comparison to theoretically proposed devices, should be helpful to people in the field, theorists and experimentalists alike.
I am however skeptical of the claimed findings of the paper. While the data is presented well and the plots are well organized, I was confused by the interpretation of the data. While reading the paper, I was constantly asking myself what type of transport mechanism the authors are imagining to occur in their device. One could imagine different transport scenarios for this three dot setup:
a) All tunnel couplings are relatively weak so that states in the dots are sharp.
In a simple noninteracting picture, transport can only occur if the bias voltage is above a certain threshold and if the levels of the different dots are tuned to be in resonance such that electrons can hop from one dot to the next. Of course there could be additional effects such as cotunneling, which could relax this criterion.
b) Tunneling to the leads is weak, but tunneling between the dots is strong. In this case the dot states are delocalized and one is probing effectively only a single dot.
c) One lead is somewhat strongly coupled to the adjacent dot.
In this case, the lead is effectively larger and one is really only probing resonances in one or two dots (depending on the dot-dot tunneling).
The authors never really clarify which regime they believe their device to be in. Only at the very end they make a statement
"We also show in supplementary materials that the tunnel barrier located at the left end near N1 is significantly more open compared to the right tunnel barrier at N2. The inter-dot barriers are also low."
This statement seems extremely important to me, since it facilitates the interpretation of the data. This finding (which is unfortunately only included in the SM, I think it should be part of the main text and should appear there much earlier) suggests to me that option c) is realized in this experiment and one should interpret the data in the following way: dot 1 and 2 are sufficiently well coupled to N1 so that this experiment is effectively two normal leads coupled to a single dot (dot 3). Of course there are resonances in dot 1 and 2 (one can see them in Fig 3a), but they are sufficiently broad, that their exact location in energy does not significantly affect transport.
Of course, I am open to the possibility that my viewpoint is oversimplified and gates 1 and 2 also have an effect. I just think that the evidence presented in the paper does not contradict the picture that all features can be explained in a simple model with a single dot sandwiched between two normal leads.
For instance, I am skeptical of the argument that the rather narrow features at low bias voltage above 0.2T in Fig 5a are states associated with dots 1 and 2. This seems to be in contradiction with the observation in the supplementary material that transport features are very broad when the current is flowing through dot 1 and 2, which means I would not expect sharp features. Also I think the fact that Fig 5b, with a different current path that does not pass dot 1 and 2, is more broadened does not allow one to conclude that the features are gone. They are relatively weak features in Fig 5a, so one would expect them to be invisible in a broadened Fig 5b, even if they belong to states in dot 3.
In summary, I do not see transport features in the data that can only be explained in terms of Andreev states localized in dots 1 and 2 (except for the very weak modulation in Fig 3a, in which gate 3 is tuned a particularly sensitive point close to a resonance of the third dot). I can see even less support for the claim that those states can be tuned by magnetic field.
From abstract, introduction, and conclusion, however, I would have expected this to be the case. See, e.g., the following quotes1:
"At zero magnetic field, Andreev bound states exist in all three dots. Andreev bound states from each dot are gate-dependent."
"In our device Andreev states in one of the dots reach zero energy at a lower field than in other two"
If my interpretation is correct and all features indeed originate from the third dot, then the findings in this paper would be much less interesting. Nevertheless it may be worth publishing these results in a more specialized journal. In this case an explanation of the transport mechanism needs to be included and the interpretation of the data needs adjustment. Unfortunately, I cannot recommend this article in its present form for publication in SciPost Physics.
If my interpretation can be falsified by additional data, I am willing to reconsider this assessment.
But even then there is a fundamental point I do not understand about this experiment.
If really one of the states in dot 1 or 2 would cross zero energy as a function of magnetic field, why would this show up as a zero bias peak in a three dot setup? If the third dot, the one with the smallest couplings to the leads, was detuned from zero, there should be no peak at zero bias. In other words, any resonance in a three dot setup can be interpreted either as a resonance of the dot with the weakest coupling in case two dots have a sufficiently strong coupling to the leads, or as resonances of multiple dots that need to be aligned, in case that multiple dots have a weak coupling to the leads. Hence, end-to-end transport measurements are no the best way to analyze the individual dots in multi-dot chains.
Here are some additional suggestions:
One possibility to prove that certain states belong to dot 1 and 2 is to use different current paths, e.g., from N1 to S1, S1 to S2, and so on. I understand that the broadening does not permit to gain much information from them, but the broadening itself is pretty interesting as I have discussed above. Why couldn't one study such an alternative setup as a function of any of the barrier gate voltages? The dI/dV in Fig S10a is two orders of magnitude above what is measured in other figures, so there seems to be some room to increase barrier heights without suppressing the current beyond what is measurable.
I don't understand the following statement:
"In Figs. 2(a,b), Andreev bound states appear more flat. We interpret this as D1 and D2 being stronger coupled to leads S1 and S2."
I would rather think that the strong broadening of states in 1 and 2 means, one is effectively only probing the resonance in dot 3.
The sentence
"In our device, transport is dominated by one of the quantum dots in the chain, D3 ,"
is unclear. Are the authors trying to say, that dot 3 has the weakest couplings to the leads and hence all visible resonances should be interpreted as originating in the third dot?
On grammar: there are a number of indefinite articles missing throughout the paper, please check. One example on p.3:
For quantum dot chain with hard induced gap, zero-bias peak only occurs
-> For a quantum dot chain with a hard induced gap, a zero-bias peak only occurs
What does the statement on p 3:
" If we set Vd3=365mV, we have zero-bias peaks on both nanowire ends."
mean? You only probe end-to-end transport, so all features are associated with both ends.

---

## Editorial Decision

unknown